# Latent Collaboration in Multi-Agent Systems

**Jiaru Zou**[1 2 * †] **Ruizhong Qiu**[3 †] **Gaotang Li**[3 †] **Xiyuan Yang**[3 †] **Katherine Tieu**[3 †] **Pan Lu**[2 †] **Ke Shen**
**Hanghang Tong**[3] **Yejin Choi**[2] **Jingrui He**[3 ✉] **James Zou**[2 ✉] **Mengdi Wang**[1 ✉] **Ling Yang**[1 ✉]

Project Page: https://github.com/Gen-Verse/LatentMAS

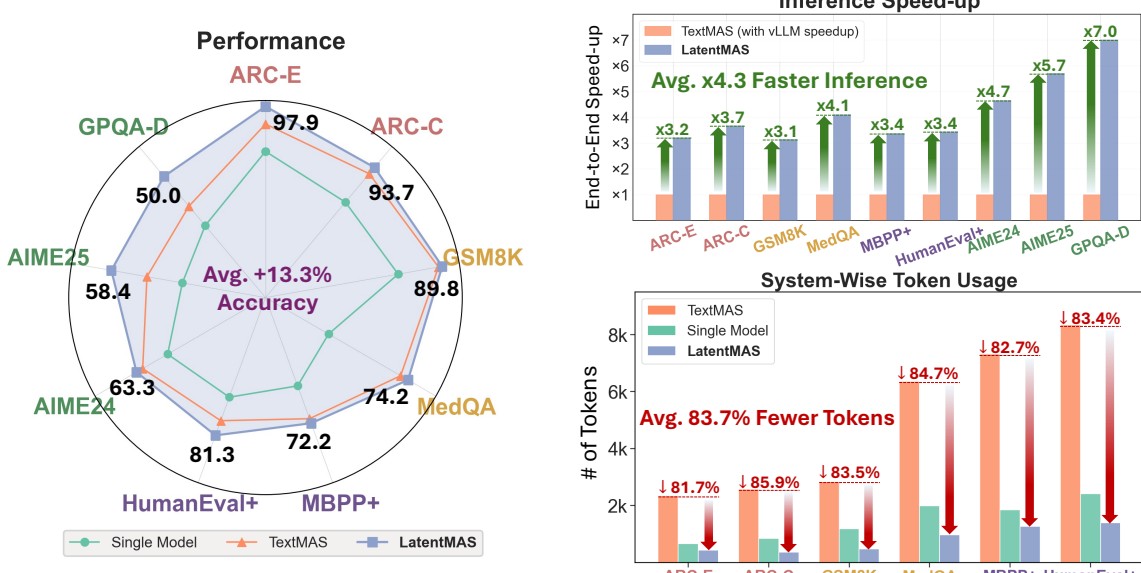

*Figure 1.* Performance of LatentMAS across (i) task accuracy (%), (ii) inference speed (time(s) / run), and (iii) token usage (per token) over 9 benchmarks and 3 LLM model scales under the Hierarchical MAS setting. LatentMAS consistently improves system-level reasoning accuracy while substantially reducing computational overhead compared with single model and text-based MAS.

## Abstract

Multi-agent systems (MAS) extend large language models (LLMs) from independent single-model reasoning to coordinative system-level intelligence. While existing LLM agents depend on text-based mediation for reasoning and communication, we take a step forward by enabling models to collaborate directly within the continuous latent space. We introduce **LatentMAS**, an *end-to-end training-free* framework that enables pure latent collaboration among LLM agents. In LatentMAS, each agent first performs auto-regressive latent thoughts generation through last-layer hidden embeddings instead of text. Then, a shared latent working memory preserves and transfers each agent's internal representations and latent thoughts, ensuring lossless information exchange without re-encoding. We provide detailed theoretical analyses showing that LatentMAS achieves higher expressiveness and lossless information preservation with lower overall complexity than standard text-based MAS. In addition, empirical evaluations across 9 comprehensive benchmarks spanning math and science reasoning, commonsense understanding, and code generation show that LatentMAS outperforms advanced single agents and text-based MAS baselines, achieving up to 14.6% higher accuracy, reducing output token usage by 70.8%-83.7%, and providing 4×-4.3× faster end-to-end inference.

*Project Lead. †Core Contributors. [1]Princeton University [2]Stanford University [3]UIUC. Correspondence to: <jingrui@illinois.edu>, <jamesz@stanford.edu>, <mengdiw@princeton.edu>, <ly1988@princeton.edu>.

*Proceedings of the 43rd International Conference on Machine Learning*, Seoul, South Korea. PMLR 306, 2026. Copyright 2026 by the author(s).

## 1. Introduction

Model collaboration emerges as the foundation of system-level intelligence in the era of Agentic AI (Acharya et al., 2025). Recent advances in multi-agent systems (MAS) (Wu et al., 2024; Hu et al., 2025) have catalyzed a paradigm shift from solitary, model-centric reasoning into a collabora-

tive endeavor among multiple interacting models. Among these, large language model (LLM)-based MAS has been adopted across various downstream applications, including cooperative math and science reasoning (Pezeshkpour et al., 2024; Zhou et al., 2025), automated code generation (Yang et al., 2024b; Ding et al., 2025), and open-domain question answering (Jin et al., 2025; Li et al., 2025c; 2026). Within LLM-based MAS, natural language or text generally serves as the *lingua franca* – the common medium that carries each agent's internal thoughts and enables communication across different agents (Guo et al., 2024).

Beyond explicit text, several studies have explored the use of LLMs' continuous latent space as a new form of "model language," (Chen et al., 2025b) by either (i) leveraging hidden representations within transformers to enable single model's internal latent chain-of-thought (CoT) reasoning (Hao et al., 2024; Zheng et al., 2025; Zhang et al., 2025), or (ii) employing KV caches or layer embeddings for information exchange across two models (Liu et al., 2024; Fu et al., 2025). However, a comprehensive model collaboration framework unifying both latent reasoning and latent communication remains unexplored. Moving one step forward, we aim to investigate:

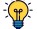 *Can MAS achieve pure latent collaboration?*

To address this question, we introduce **LatentMAS**, an end-to-end collaborative framework that operates entirely within the continuous latent space. Our core design integrates both internal *latent thoughts generation* and cross-agent *latent working memory transfer*. Inside each agent, reasoning unfolds through auto-regressive generation of last-layer hidden representations, capturing the model's ongoing internal thoughts without explicit decoding. Across agents, information is exchanged via shared latent working memory stored in layer-wise KV caches, capturing both the input context and newly generated latent thoughts. Overall, LatentMAS is completely *training-free*, enabling all agents to think and interact purely through their internal latent representations.

Building on this framework design, LatentMAS is grounded on three foundational principles, which are substantiated through comprehensive theoretical and empirical analyses:

- **Reasoning Expressiveness:** Hidden representations naturally encode model continuous thoughts, making each latent step more expressive than discrete tokens and capable of conveying richer semantic information.

- **Communication Fidelity:** Latent working memory preserves input representations and latent thoughts, enabling lossless information transfer across agents.

- **Collaboration Complexity:** LatentMAS exhibits higher collaborative expressiveness than text-based MAS with substantially lower inference complexity.

The first two principles jointly underscore the advantage of LatentMAS by enabling richer latent reasoning and lossless latent communication. The last principle characterizes how latent collaboration reshapes the efficiency-expressiveness tradeoff, illustrating that LatentMAS enables scalable multi-agent coordination with reduced computational overhead.

To empirically assess the efficacy of LatentMAS, we conduct comprehensive evaluations on 9 benchmarks spanning math and science reasoning, commonsense understanding, and code generation (Figure 1). Across both sequential and hierarchical MAS settings, and over five different backbone models spanning the Qwen3 (Yang et al., 2025) and Llama3 (Grattafiori et al., 2024) families, LatentMAS consistently outperforms advanced single-model and text-based MAS baselines by (i) improving accuracy by up to 14.6%, (ii) reducing output token usage by 70.8%-83.7%, and (iii) delivering 4×-4.3× faster end-to-end inference. These results demonstrate that latent collaboration not only enhances system-level reasoning quality but also provides consistent efficiency gains without any additional training. Further detailed analyses of latent thought expressiveness, working-memory transfer, and input-output alignment confirm that LatentMAS enables semantically meaningful, lossless, and stable collaboration entirely in latent space.

## 2. Preliminary and Notations

**Auto-regressive Generation in Transformer.** Let $f_\theta(\cdot)$ denotes the function computed by a standard Transformer model (Vaswani et al., 2017), parameterized by $\theta$. Given an input sequence $x = (x_1, x_2, \ldots, x_T)$, the transformer $f_\theta(\cdot)$ first encodes each token via its input embedding layer $W_{\text{in}}$ to obtain token embeddings up to step $t$, i.e., $E = [e_1, e_2, \ldots, e_t] \in \mathbb{R}^{t \times d_h}$, where $d_h$ is the model's hidden dimension. The input token embeddings $E$ then successively process through $L$ transformer layers in the forward pass through the model's residual stream, yielding the final-layer hidden representations $H = [h_1, h_2, \ldots, h_t] \in \mathbb{R}^{t \times d_h}$. For next token generation, the model computes:

$$f_\theta(x_{t+1} \mid x_{\leq t}) = \text{softmax}(h_t W_{\text{out}}), \tag{1}$$

where $W_{\text{out}}$ denotes the language model head that maps the hidden representation to the vocabulary space. Each token is generated in an auto-regressive manner and appended to the input sequence. For *latent generation*, the model performs auto-regression over hidden representations instead of tokens, bypassing explicit decoding (Hao et al., 2024).

**KV Cache as Working Memory.** In decoder-only Transformers, the Key-Value (KV) cache functions as a dynamic working memory during auto-regressive generation, storing intermediate representations from previous decoding steps to avoid redundant computation. Specifically, given the input embeddings $E$, each transformer layer projects them through projection matrices $W_Q, W_K, W_V$ to obtain

$Q, K, V$. When the next token at step $t + 1$ is generated, the model appends its embedding to the input sequence and updates the cache $(K_{\text{cache}}, V_{\text{cache}})$ as:

$$K_{\text{cache}} \leftarrow [K_{\leq t}; K_{t+1}], \quad V_{\text{cache}} \leftarrow [V_{\leq t}; V_{t+1}], \quad (2)$$

where $K_{\leq t}$, $V_{\leq t}$ are accumulated key/value matrices from all previous steps and $K_{t+1}$, $V_{t+1}$ are new key/value vectors computed from the current token's hidden state. This accumulative property enables the KV cache to maintain a growing working memory of model internal representations.

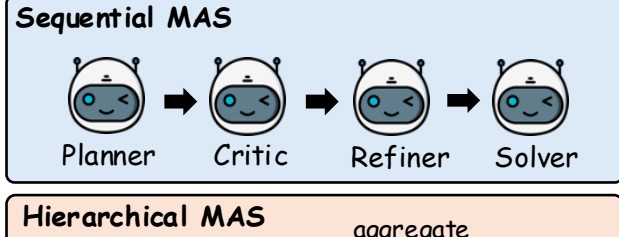

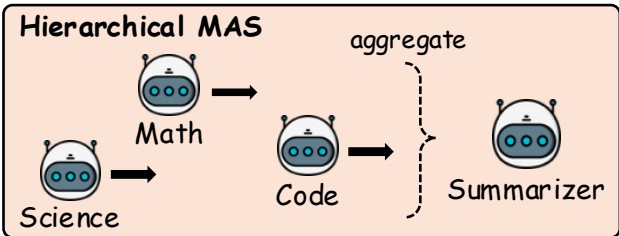

*Figure 2.* Illustration of sequential and hierarchical MAS settings.

**LLM-based MAS Setting.** We consider a multi-agent system $\mathcal{S}$ composed of $N$ agents, denoted as $\mathcal{A} = \{A_1, A_2, \ldots, A_N\}$, where each agent $A_i$ is an LLM corresponding to $f_{\theta_i}$ above. At inference time, an input question $q$ is provided to the system $\mathcal{S}$, which orchestrates interactions among agents to collaboratively produce a final answer $a$ corresponding to $q$. As MAS design paradigms are not definitive in general and often vary across downstream tasks (Tran et al., 2025; Cemri et al., 2025), we do not restrict our latent collaboration design to any particular architecture. Instead, we adopt two most commonly used MAS settings (*sequential* and *hierarchical*) as the bases to experimentally evaluate our method. Figure 2 illustrates the two MAS architecture settings. In the **sequential MAS**, we adopt a chain-of-agents design (Zhang et al., 2024b; Zhao et al., 2026) comprising four LLM agents: `planner`, `critic`, `refiner`, and `solver`. These agents assume complementary reasoning roles and are organized in a sequential pipeline, where the CoT output of each agent with the question $q$ serves as the input to the next agent. In the **hierarchical MAS**, we adopt a domain-specialized design (Zhuge et al., 2024; Zhao et al., 2025). Multiple LLM agents, including `code`, `math`, and `science` agents, operate as different domain experts. Each agent independently reasons over the question $q$ from its disciplinary perspective. A `summarizer` agent then receives all intermediate responses along with the question $q$ and performs hierarchical aggregation to synthesize and refine the final answer.

# 3. LatentMAS

We introduce LatentMAS, an end-to-end latent collaboration framework that all agents reason and communicate entirely within the latent space. As illustrated in Figure 3, our method enables LLM agents to (i) auto-regressively generate expressive latent thoughts (Section 3.1), (ii) transfer latent working memory across agents with lossless communication fidelity (Section 3.2), and (iii) achieve lower inference complexity than text-based MAS while supporting a higher degree of model coordination (Section 3.3).

**Method Roadmap.** In the following sub-sections, we present the complete pipeline of LatentMAS, detailing each component and interleaving theoretical analyses to justify the corresponding design principles.

### 3.1. Auto-regressive Latent Thoughts Generation

We first describe how each LLM agent performs latent reasoning through its layer-wise hidden states. Instead of decoding explicit tokens, reasoning unfolds directly within the agent by auto-regressively appending hidden representations from the final model layer.

Given the input embeddings $E = [e_1, e_2, \ldots, e_t]$ containing the information from the question $q$ and each agent's instruction prompt, each LLM agent $A_i \in \mathcal{A}$ passes $E$ through $L$ transformer layers to compute the last-layer hidden representation $h_t$ at current step $t$. Then, we insert $h_t$ as the input embedding for the next step $t + 1$, replacing the original decoding and next-token embedding processes used in standard token generation. We auto-regressively repeat the process for $m$ latent steps, yielding a sequence of newly generated last-layer hidden states $H = [h_{t+1}, h_{t+2}, \ldots, h_{t+m}]$. We define the continuous output representations $H$ as the *latent thoughts* generated by $A_i$.

**Input-Output Distribution Alignment.** Since the newly generated $H$ form a sequence of dense, high-level representations, directly inserting them into shallow layers as input embeddings may lead to out-of-distribution activations (Meegahapola et al., 2019; Zhou et al., 2019), as these hidden states differ from the statistical patterns of learned token embeddings. To mitigate this in a training-free manner, we propose a *linear alignment operator* that maps last-layer hidden states back to the valid input embeddings. Specifically, given $W_{\text{in}}$, $W_{\text{out}}$ as the input and output embedding layers of $A_i$, we seek a projection matrix $W_a \in \mathbb{R}^{d_h \times d_h}$ that maps each output vector $h \in H$ to a new input vector $e$ to align with valid input space defined by $W_{\text{in}}$:

$$e = h W_a, \quad \text{where } W_a \approx W_{\text{out}}^{\dagger} W_{\text{in}}. \quad (3)$$

Here, $W_{\text{out}}^{\dagger}$ is the pseudo-inverse (Hoerl & Kennard, 1970) of $W_{\text{out}}$ (in Appendix A, we provide a detailed theoretical justification for the effectiveness of $W_a$ in the input–output

## Latent Collaboration (LatentMAS)

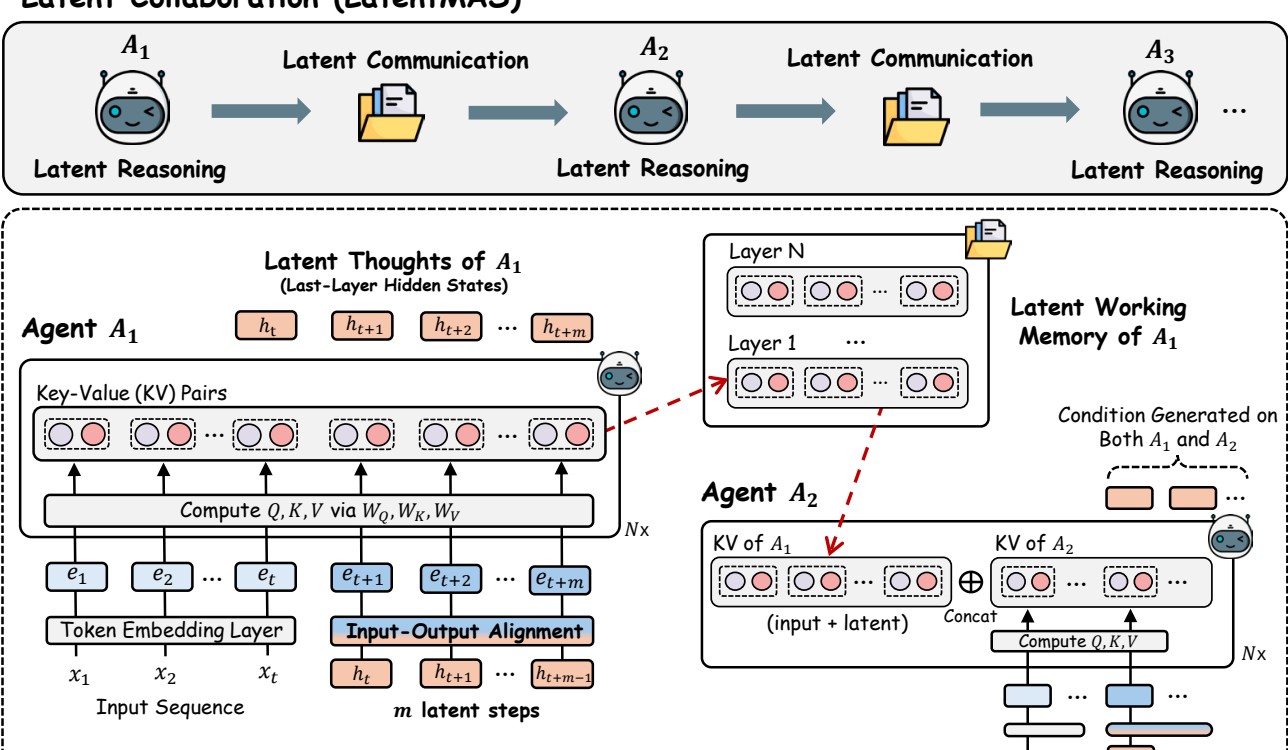

*Figure 3.* **Overview of the LatentMAS framework.** To enable end-to-end system-wide latent collaboration, each LLM agent in LatentMAS first performs reasoning in latent space by generating latent thoughts from last-layer hidden states (**Section 3.1**), and then communicates its internal reasoning to subsequent agents via layer-wise transfer through shared latent working memory (**Section 3.2**).

alignment process, followed by a formal derivation of $W_a$). We then append the aligned vector $e$ into the input sequence for auto-regressive latent generation. Note that $W_a$ is a small projection matrix of size $d_h \times d_h$ (e.g., $d_h$=1024 for Qwen3-0.6B) and is computed once and reused in all subsequent latent steps. This design makes the alignment computationally negligible while maintaining distributional consistency between latent and discrete representations.

**Expressiveness on Continuous Latent Thoughts.** With the mechanism of latent thought generation established within each agent, we next provide a theoretical analysis to quantify its representational advantage over conventional discrete token generation. The following theorem formalizes that latent thoughts, which inherently preserve richer semantic structures, achieve substantially higher expressive capacity than discrete text-based reasoning.

**Theorem 3.1** (**Expressiveness of Latent Thoughts**).
*Under the Linear Representation Hypothesis on $h$ (stated in Assumption B.1), if the sequence of all latent thoughts with length $m$ can be expressed losslessly through corresponding text-based reasoning, then the length of text (in tokens) needs to be at least $\Omega\left(d_h m / \log|\mathcal{V}|\right)$, where $|\mathcal{V}| > 1$ denotes the vocabulary size.*

*Remark* 3.2. Theorem 3.1 suggests that latent thoughts generation can be $O\left(d_h / \log|\mathcal{V}|\right)$ times more efficient than text-based reasoning. In addition, the expressiveness scales linearly with $d_h$, implying that larger models inherently exhibits greater latent reasoning capacity.

As an illustration to Remark 3.2, for Qwen3-4B / 8B / 14B models (Yang et al., 2025), latent thoughts generation can be 235.7 / 377.1 / 471.4 times more efficient than text-based reasoning. The full proof of Theorem 3.1 is provided in Appendix B.1. Beyond reasoning within individual agents, collaboration in LatentMAS further relies on how these agents exchange latent information, which we detail next.

### 3.2. Thoughts Transfer via Latent Working Memory

In text-based MAS, after one LLM agent completes its generation, the natural language output is directly appended to the input sequence of the next agent. However, since each agent in LatentMAS performs hidden-state generation without explicit text outputs, we design a new *latent working memory* transfer mechanism to ensure lossless information preservation and exchange.

For clarity, we describe the memory transfer mechanism using the first two consecutive LLM agents $A_1, A_2 \in \mathcal{A}$. As shown in Figure 3, agent $A_1$ first performs $m$ latent steps

of generation (Section 3.1). After completing these steps, we extract the KV-caches from all $L$ transformer layers of $A_1$ once and collect the *latent working memory* of $A_1$ as:

$$\mathcal{M}_{A_1} = \left\{ \left( K_{A_1,\text{cache}}^{(l)}, V_{A_1,\text{cache}}^{(l)} \right) \Big| l = 1, 2, \ldots, L \right\},$$

$$\text{with } K_{A_1,\text{cache}}^{(l)} = [K_{A_1,1}^{(l)}, \ldots, K_{A_1,t+m}^{(l)}], \qquad (4)$$

$$\text{and } V_{A_1,\text{cache}}^{(l)} = [V_{A_1,1}^{(l)}, \ldots, V_{A_1,t+m}^{(l)}].$$

Here $K_{A_1,\text{cache}}^{(l)}$ and $V_{A_1,\text{cache}}^{(l)}$ are accumulated key and value matrices at the $l$-th layer. Unlike direct cache-sharing methods that exchange information mainly on prefilled input context across models, the collection of layer-wise caches in $\mathcal{M}_{A_1}$ encapsulates both the initial input context and the newly generated latent thoughts of agent $A_1$.

Next, the successive agent $A_2$ integrates the working memory $\mathcal{M}_{A_1}$ from agent $A_1$. Before $A_2$ generates latent thoughts (i.e., last-layer hidden states), we perform layer-wise concatenation to update its KV cache by prepending each $K_{A_1,\text{cache}}^{(l)}$ and $V_{A_1,\text{cache}}^{(l)}$ to existing $K_{A_2,\text{cache}}^{(l)}$ and $V_{A_2,\text{cache}}^{(l)}$. By doing so, the new latent thoughts generation in $A_2$ is conditioned on both the working memory of $A_1$ and its own internal representations.

**Lossless Information Transfer.** The latent working memory transfer mechanism ensures that each succeeding agent in LatentMAS seamlessly receives its predecessor's complete output without re-encoding. In the following theorem, we further formalize this property to demonstrate that latent working memory transfer guarantees information fidelity equivalent to explicit input exchange.

> **Theorem 3.3** (**Information Preservation via Latent Working Memory**). *In both latent and text-based reasoning, the outputs of an agent when receiving latent working memory from preceding agents are equivalent to those obtained when directly inputting the preceding agents' outputs.*

Full proof of Theorem 3.3 is provided in B.2. With lossless information preservation, we transfer latent working memory in KV rather than directly transmitting hidden states to avoid redundant recomputation for the successive agent.

### 3.3. End-to-End Pipeline with Complexity Analyses

For the remaining agents in LatentMAS, we follow the same latent thoughts generation and working memory transfer mechanism described above. Specifically, agent $A_3$ inherits the working memory $\mathcal{M}_{A_2}$ from the preceding agent $A_2$, performs auto-regressive last-layer hidden state generation, and subsequently transmits its updated latent working memory $\mathcal{M}_{A_3}$ to the next agent. This process continues across all agents in LatentMAS, with only the last agent decoding the final answer in texts. In the following theorem, we theoretically analyze the overall complexity of LatentMAS.

> **Theorem 3.4** (**LatentMAS Complexity**). *The time complexity for each agent of LatentMAS is $O\big((d_h^2 m + d_h m^2 + d_h tm)L\big)$, where $t$ is the input length of this agent, and $m$ is the length of latent thoughts. In contrast, assuming Theorem 3.1, the time complexity for each agent of the vanilla text-based MAS needs to be $O\big((d_h^3 m \frac{1}{\log |\mathcal{V}|} + d_h^3 m^2 \frac{1}{\log^2 |\mathcal{V}|} + d_h^2 tm \frac{1}{\log |\mathcal{V}|})L + d_h^2 |\mathcal{V}| m \frac{1}{\log |\mathcal{V}|}\big)$ to achieve the same expressiveness.*

The full proof of Theorem 3.4 is provided in B.3. It is worth noting that LatentMAS is agnostic to specific model collaboration strategies and can be seamlessly applied to sequential, hierarchical, or other advanced MAS designs.

## 4. Empirical Evaluations

**Tasks and Datasets.** We conduct comprehensive evaluations of LatentMAS across 9 benchmarks spanning both general-purpose and reasoning-intensive tasks: (i) *Math & Science Reasoning*, including GSM8K (Cobbe et al., 2021), AIME24 (Maxwell-Jia, 2024), AIME25 (math ai, 2025), GPQA-Diamond (Rein et al., 2023), and MedQA (Yang et al., 2024a); (ii) *Commonsense Reasoning*, including ARC-Easy (Clark et al., 2018b) and ARC-Challenge (Clark et al., 2018a); and (iii) *Code Generation*, including MBPP-Plus (Liu et al., 2023) and HumanEval-Plus (Liu et al., 2023). Additional benchmark details are provided in Appendix C.1.

**Models and Baselines.** We adopt 5 off-the-shelf models from both the Qwen3 family (Yang et al., 2025) (4B, 8B, and 14B) and the Llama 3 family (Grattafiori et al., 2024) (3B, 8B) to construct LatentMAS at different scales. For baseline comparison, we evaluate LatentMAS against: (i) *Single LLM agents (Single)*, where a single LLM directly performs standard auto-regressive generation with token-level decoding; (ii) *Sequential text-based MAS (Sequential TextMAS)*, following the chain-of-agents design (Zhang et al., 2024b) with text-mediated reasoning and communication; and (iii) *Hierarchical text-based MAS (Hierarchical TextMAS)*, where domain-specialized agents collaborate through a summarizer (Zhuge et al., 2024) using text-based reasoning and communication. Detailed model and baseline implementations are provided in Appendix C.2.

**Implementation Details.** For latent thoughts generation, we compute the realignment matrix $W_a$ once per run and reuse it across all inference steps. Each LLM agent performs $m \in \{0, 10, 20, 40, 80\}$ latent steps during reasoning. For working memory transfer, we directly concatenate the KV caches from the immediately preceding agent into the corresponding transformer layers through the `past_key_values` interface in HuggingFace `Transformers` (Face, 2025). Besides the HuggingFace implementation, we also integrate all baseline methods and LatentMAS with the `vLLM` backend (Kwon et al., 2023),

*Table 1.* **Main results of LatentMAS on 6 general tasks under the Sequential MAS setting.** We report 3 metrics in total, including task accuracy (%, **"Acc."**), total output token usage (**"Token"**), and end-to-end inference speed (time(s) / run, **"Speed"**). We compare LatentMAS with both TextMAS and single-model ("Single") baselines. For each metric, we **bold** the better performance and visualize LatentMAS gains over TextMAS in the █Improve█ columns. Additional results under the Hierarchical MAS setting are shown in Table 3.

| Tasks | Metrics | Qwen3-4B | | | Improve | Qwen3-8B | | | Improve | Qwen3-14B | | | Improve |
|---|---|---|---|---|---|---|---|---|---|---|---|---|---|
| | | Single | TextMAS | LatentMAS | | Single | TextMAS | LatentMAS | | Single | TextMAS | LatentMAS | |
| *Sequential MAS Setting* | | | | | | | | | | | | | |
| **ARC-E** | Acc. | 95.4 | 96.4 | **98.6** | ↑ **2.2** | 95.6 | 99.1 | 98.8 | ↓ 0.3 | 97.2 | 99.0 | **99.4** | ↑ **0.4** |
| | Token | 724 | 2420 | **581** | ↓ **76.0%** | 656 | 2085 | **490** | ↓ **76.5%** | 608 | 1670 | **224** | ↓ **86.6%** |
| | Speed | 369 | 2874 | **512** | ×**5.6** | 404 | 3702 | **1759** | ×**2.1** | 551 | 9171 | **2124** | ×**4.3** |
| **ARC-C** | Acc. | 89.2 | 90.0 | **92.3** | ↑ **2.3** | 91.0 | 94.6 | 94.4 | ↓ 0.2 | 92.6 | 95.9 | 95.6 | ↓ 0.3 |
| | Token | 913 | 2678 | **718** | ↓ **73.2%** | 846 | 2252 | **529** | ↓ **76.5%** | 773 | 2985 | **426** | ↓ **85.7%** |
| | Speed | 97 | 1579 | **260** | ×**6.1** | 266 | 2059 | **703** | ×**2.9** | 338 | 5125 | **1136** | ×**4.5** |
| **GSM8K** | Acc. | 82.4 | 89.8 | 88.2 | ↓ 1.6 | 81.1 | 92.3 | **93.8** | ↑ **1.5** | 83.7 | 93.8 | **95.2** | ↑ **1.4** |
| | Token | 1136 | 3172 | **607** | ↓ **80.9%** | 1280 | 2324 | **860** | ↓ **63.0%** | 1118 | 3324 | **644** | ↓ **80.6%** |
| | Speed | 469 | 1970 | **375** | ×**5.3** | 449 | 1739 | **543** | ×**3.2** | 536 | 3729 | **1952** | ×**1.9** |
| **MedQA** | Acc. | 47.7 | 65.3 | **66.3** | ↑ **1.0** | 53.0 | 75.0 | **75.3** | ↑ **0.3** | 64.7 | 80.3 | **80.7** | ↑ **0.4** |
| | Token | 2134 | 3962 | **1685** | ↓ **57.5%** | 2098 | 4260 | **1555** | ↓ **63.5%** | 1746 | 3444 | **1841** | ↓ **46.5%** |
| | Speed | 236 | 1267 | **438** | ×**2.9** | 476 | 1923 | **928** | ×**2.1** | 1360 | 4142 | **1420** | ×**2.9** |
| **MBPP+** | Acc. | 63.5 | 69.8 | **73.5** | ↑ **3.7** | 64.8 | 69.5 | **74.6** | ↑ **5.1** | 68.5 | 72.8 | **75.7** | ↑ **2.9** |
| | Token | 1634 | 4420 | **1339** | ↓ **69.7%** | 2053 | 3695 | **1164** | ↓ **68.5%** | 1858 | 4971 | **1621** | ↓ **67.4%** |
| | Speed | 523 | 2148 | **577** | ×**3.7** | 1064 | 3628 | **1275** | ×**2.8** | 2410 | 8728 | **2400** | ×**3.6** |
| **HumanEval+** | Acc. | 75.0 | 79.7 | **79.9** | ↑ **0.2** | 74.4 | 80.5 | **80.5** | ↑ 0.0 | 76.8 | 81.1 | **86.5** | ↑ **5.4** |
| | Token | 2380 | 5987 | **1775** | ↓ **70.4%** | 2507 | 4593 | **1866** | ↓ **59.4%** | 2366 | 5934 | **2042** | ↓ **65.6%** |
| | Speed | 274 | 1044 | **350** | ×**3.0** | 502 | 1619 | **497** | ×**3.3** | 1084 | 4062 | **1285** | ×**3.2** |

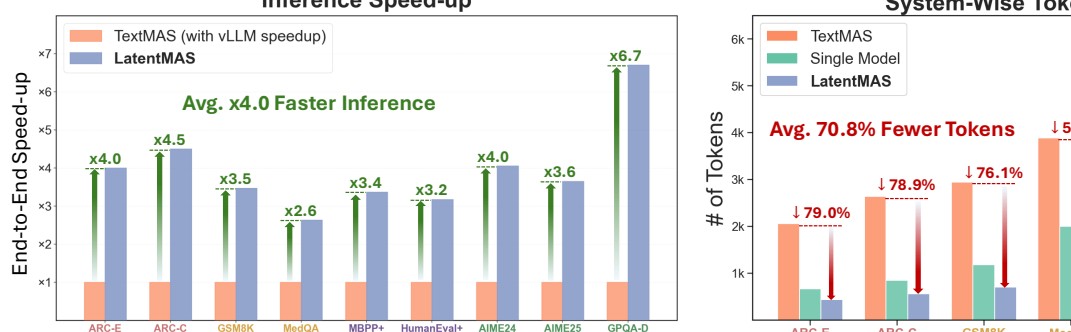

*Figure 4.* **Efficiency gains of LatentMAS over single model and TextMAS under the Sequential MAS setting.** *Left:* LatentMAS achieves consistent faster end-to-end inference speed. *Right:* LatentMAS requires fewer system-wise token usage in total.

enabling prefix caching and tensor-parallel inference for efficient deployment of larger LLM agents. We perform hyperparameter tuning and report the mean performance over three independent runs. Across both baselines and our method, we set all LLM agents with a temperature of 0.6 and a top-$p$ of 0.95. We adjust the maximum output length for each task according to its relative difficulty. We set the maximum length to 2,048 tokens for ARC-Eacy, ARC-Challenge, and GSM8K, 4096 tokens for MedQA, MBPP+, and Humaneval+, 8,192 tokens for GPQA and 20,000 tokens for AIME24 and AIME25. All experiments are conducted on 8×NVIDIA A100-80G GPUs.

### 4.1. Main Results

Tables 1, 2, and 3 report the overall performance of Latent-MAS across 9 general and reasoning-intensive benchmarks

built from 5 different scales of LLM backbones. To thoroughly examine collaboration behaviors during inference, we evaluate each method from three complementary perspectives: (i) *task accuracy*, (ii) *system throughput (total output tokens)*, and (iii) *end-to-end inference speed*. Across all tasks, LatentMAS consistently improves over the single-model baseline by an average of 14.6% and 13.3% under the sequential and hierarchical settings, respectively, and further yields gains of 2.8% and 4.6% over text-based MAS. Under identical MAS architectures, LatentMAS provides 4× and 4.3× faster inference speed on average compared with sequential and hierarchical text-based MAS. Additionally, as the agents' collaboration occurs entirely in latent space, LatentMAS reduces token usage by 70.8% and 83.7% relative to sequential and hierarchical TextMAS. We leave additional results on Llama backbones in Appendix D.2.

*Table 2.* **Main results of LatentMAS on 3 reasoning-intensive tasks under Sequential and Hierarchical MAS settings.** We report accuracy, token usage, and end-to-end speed, and highlight the performance gains following the same evaluation protocol as in Table 1.

| Tasks | Metrics | Qwen3-8B | | | Improve | Qwen3-14B | | | Improve |
|---|---|---|---|---|---|---|---|---|---|
| | | Single | TextMAS | LatentMAS | | Single | TextMAS | LatentMAS | |
| *Sequential MAS Setting* | | | | | | | | | |
| **AIME24** | Acc. | 50.0 | 53.3 | **56.7** | ↑ **3.4** | 63.3 | 63.3 | **66.7** | ↑ **3.4** |
| | Token | 12891 | 38596 | **8953** | ↓ **76.8%** | 11263 | 32092 | **10593** | ↓ **67.0%** |
| | Speed | 421 | 2808 | **688** | × **4.1** | 1018 | 4554 | **1149** | × **4.0** |
| **AIME25** | Acc. | 46.7 | 53.3 | **53.3** | ↑ 0.0 | 56.7 | 60.0 | **63.3** | ↑ **3.3** |
| | Token | 14692 | 45088 | **8699** | ↓ **80.7%** | 11298 | 44618 | **11402** | ↓ **74.4%** |
| | Speed | 450 | 3150 | **820** | × **3.8** | 1040 | 5184 | **1473** | × **3.5** |
| **GPQA-Diamond** | Acc. | 39.9 | 43.4 | **45.5** | ↑ **2.1** | 48.5 | 51.5 | **52.0** | ↑ **0.5** |
| | Token | 6435 | 17986 | **4571** | ↓ **74.6%** | 5547 | 12676 | **5454** | ↓ **57.0%** |
| | Speed | 813 | 5771 | **854** | × **6.8** | 1043 | 9714 | **1475** | × **6.6** |
| *Hierarchical MAS Setting* | | | | | | | | | |
| **AIME24** | Acc. | 50.0 | 53.3 | 53.3 | ↑ 0.0 | 63.3 | 70.0 | **73.3** | ↑ **3.3** |
| | Token | 12891 | 42629 | **7526** | ↓ **82.3%** | 11263 | 29025 | **10230** | ↓ **64.8%** |
| | Speed | 421 | 3132 | **776** | × **4.0** | 1018 | 5718 | **1089** | × **5.3** |
| **AIME25** | Acc. | 46.7 | 50.0 | 50.0 | ↑ 0.0 | 56.7 | 66.7 | 66.7 | ↑ 0.0 |
| | Token | 14692 | 53929 | **13230** | ↓ **75.5%** | 11298 | 50003 | **9527** | ↓ **80.9%** |
| | Speed | 450 | 3488 | **616** | × **5.7** | 1040 | 6019 | **1056** | × **5.7** |
| **GPQA-Diamond** | Acc. | 39.9 | 43.0 | **46.9** | ↑ **3.9** | 48.5 | 52.0 | **53.0** | ↑ **1.0** |
| | Token | 6435 | 22450 | **3395** | ↓ **84.9%** | 5547 | 20931 | **3606** | ↓ **82.8%** |
| | Speed | 813 | 6108 | **798** | × **7.7** | 1043 | 9119 | **1458** | × **6.3** |

## 4.2. Efficiency Analyses on Latent Collaboration.

**End-to-end Inference Speed.** As established earlier in Theorem 3.1, LatentMAS can achieve orders-of-magnitude higher expressiveness, thereby enabling more efficient inference than text-based MAS. We further empirically validate this advantage through comparisons with TextMAS. As shown in Figure 4 and 10 (left), even after accelerating the TextMAS baselines using the vLLM service, LatentMAS still achieves a 2.6×-7× speedup over the vLLM-optimized TextMAS. This improvement stems from the substantially reduced number of latent steps required for latent thoughts generation compared with the much larger decoding steps needed for per-token text generation. With fewer than 50 latent steps, LatentMAS attains comparable or even higher performance on reasoning-intensive tasks such as AIME 24/25, whereas TextMAS typically requires more than 20K output tokens to complete full text-based CoT trajectories.

**Token Throughput.** In addition, as illustrated in Figure 4 and 10 (right), LatentMAS reduces token usage by 59.4%-87.9% compared with TextMAS, as agents in LatentMAS communicate by directly transferring latent working memory into another agent's internal layers rather than relying on the text medium. LatentMAS also achieves 15.0%-60.3%

lower token usage than single agents. Compared with single-model reasoning, LatentMAS distributes the input question across collaborating agents, greatly reducing the burden on the final agent, which primarily aggregates preceding latent thoughts and decodes the final answer using only a small number of tokens. As a result, the entire system generates fewer output tokens while still achieving higher accuracy.

## 5. In-depth Analyses on LatentMAS

**Do Latent Thoughts Reflect Text Reasoning?** We first verify whether latent thoughts generation in LatentMAS produces meaningful and semantically expressive representations. To this end, we compare the distribution of newly generated last-layer embeddings in LatentMAS with the embeddings of token-by-token responses produced by TextMAS. Experiments are conducted on 300 MedQA questions, using 40 latent steps for LatentMAS and a 4096 max-token budget for the TextMAS baseline.

As shown in Figure 5, we highlight two key observations: (i) The last-layer embeddings from LatentMAS share nearly the same region of the embedding space with the token embeddings from TextMAS, indicating that latent thoughts encode similar semantic representations as the correct text

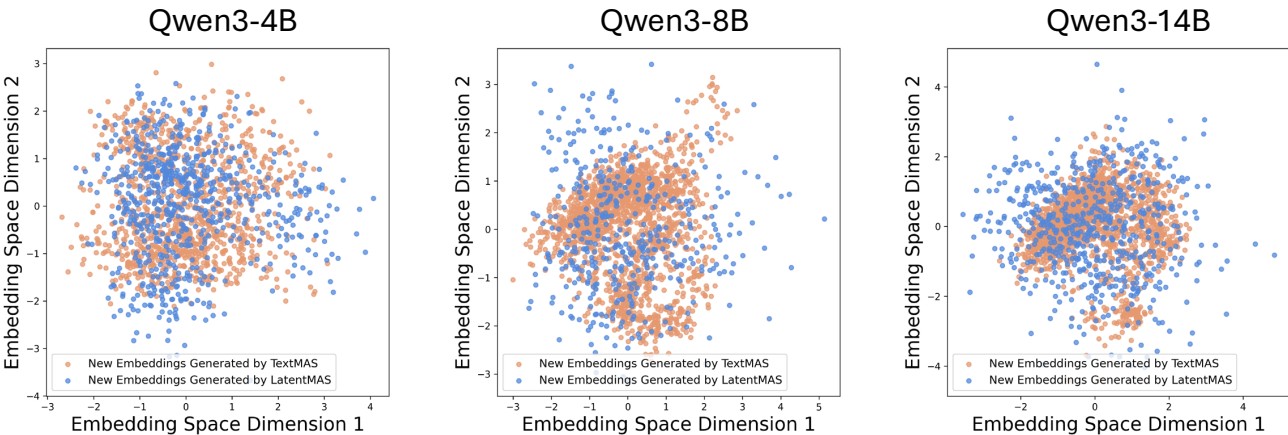

*Figure 5.* **Illustration of the semantic meaning encoded by LatentMAS.** LatentMAS's generated latent thoughts embeddings largely cover the embedding space of text-based generated tokens, indicating the semantic consistency and expressive capacity of LatentMAS.

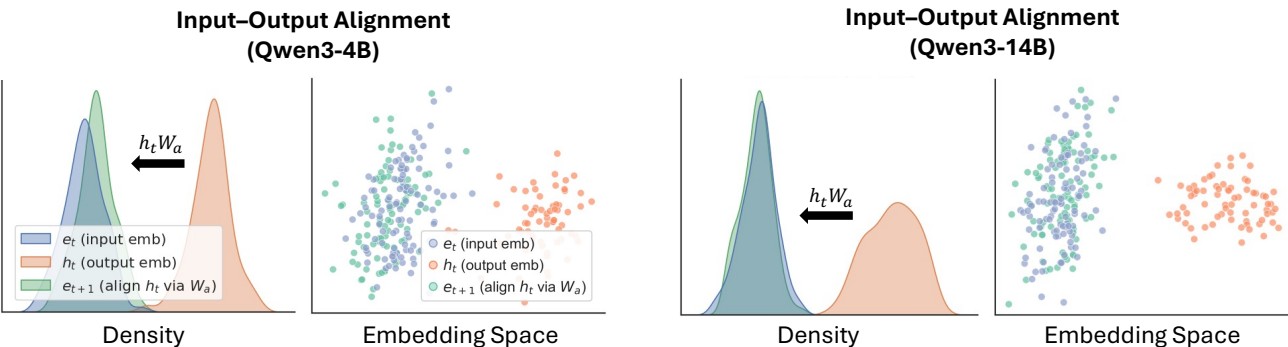

*Figure 6.* **Effectiveness of the input-output alignment $W_a$ on MedQA.** Unaligned output embeddings ($h_t$) drift away from the original input embeddings ($e_t$), while the aligned vectors ($e_{t+1}$) realign with $e_t$, demonstrating that $W_a$ preserves embedding-space structure and prevents representation drift. Analyses on additional models are provided in Appendix E.1.

responses. (ii) The last-layer embeddings from LatentMAS largely cover the distribution of token embeddings from TextMAS, indicating that latent thoughts offer greater diversity and expressive capacity than discrete tokens. Together, these findings show that latent thoughts not only capture the valid semantics of their corresponding text responses but also encode richer and more expressive representations inside. We further include a case study in Appendix J analyzing how LLM agents in LatentMAS interpret their own latent thoughts to provide additional validation.

**Effectiveness on Input-Output Alignment.** We next empirically evaluate the effectiveness of the input-output alignment in our method design. First, we compare the input vector $e_t$ obtained from the standard token embedding layer with both the newly generated output vector $h_t$ before alignment and the after-aligned vector $e_{t+1}$. As shown in Figure 6, we visualize the three embedding vectors by comparing both density distributions and geometric relationships in the projected embedding space. We observe that the new $h_t$ deviates largely from the original input embedding $e_t$. After applying $W_a$, the aligned vector $e_{t+1}$ realigns with

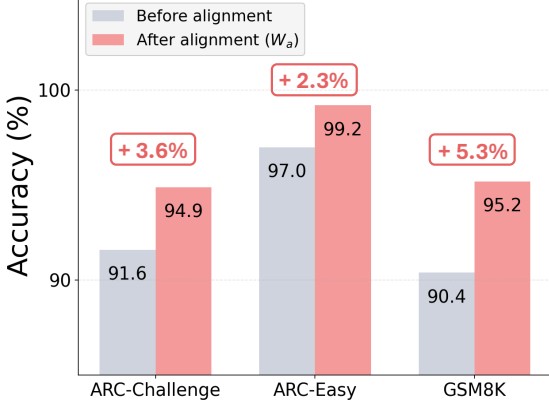

*Figure 7.* Downstream accuracy performance (%) on 3 benchmarks before/after applying the input-output alignment $W_a$.

$e_t$, indicating that $W_a$ effectively restores the geometric and statistical structure of the input embedding space and mitigates representation drift across iterative latent steps. In Figure 7, across 3 different tasks, we further compare downstream performance before and after applying $W_a$ and observe consistent accuracy gains of 2.3%-5.3% by $W_a$.

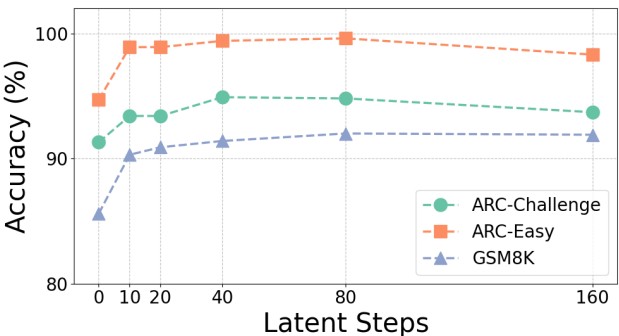

*Figure 8.* Effectiveness of different latent step depths.

**Optimal Latent Step Depth.** To understand how many latent steps are needed for optimal performance in Latent-MAS, we analyze the effect of increasing latent step depth across three downstream tasks. As shown in Figure 8, increasing the number of latent steps generally improves downstream performance, indicating that additional latent thoughts enhance collaborative expressiveness. Across the three tasks on Qwen3-14B, we find that accuracy steadily rises and peaks around 40-80 steps. Beyond this range, performance plateaus or declines, suggesting that excessive latent thought generation may introduce redundant or less useful information. Based on this observation, we adopt a moderate latent step budget within this range in practice, as it consistently provides the best accuracy-efficiency trade-off without requiring any task-specific training procedures.

**Intermediate Error Tracking within LatentMAS.** To better audit and interpret the latent collaboration process, we extend our original framework with an additional *"debug mode"* to enable a flexible conversion between inner latent thoughts and outer human-readable texts within each agent. We leave implementation details on how to converse to the debug mode in LatentMAS, together with additional detailed analyses and case studies, to Appendix E,F, and J.

## 6. Related Work

**LLM-based Multi-agent Systems.** Recent studies in Agentic AI have extended classical multi-agent systems (Hong et al., 2023; Park et al., 2023a; Yang et al., 2024c) grounded in traditional reinforcement learning and policy coordination, to modern LLM settings (Li et al., 2025b; Tan et al., 2025), enabling models to operate as autonomous agents that collaborate in reasoning, planning, and problem-solving (Tao et al., 2024; Wang et al., 2025c; Zhao et al., 2025). Early methods (Yao et al., 2022; Wu et al., 2024; Li et al., 2023) coordinate multiple LLMs through explicit dialogue and role assignment. Additional methods introduce structured communication protocols to enhance cooperation efficiency (Ye et al., 2025a) and emergent specialization (Mieczkowski et al., 2025; Huang et al., 2026) among agents. Building upon these advanced text-MAS methods, our work

aims to construct a multi-agent collaboration system entirely in latent space, treating agents as tightly integrated components to achieve more expressive and efficient coordination.

**Model Collaboration in Latent Space.** Recent studies on model collaboration (Tran et al., 2025) have gradually shifted from text-level coordination toward interaction directly in latent representations. Instead of exchanging information through texts, these works (Coda-Forno et al., 2025; Ye et al., 2025a; Fu et al., 2025; Zheng et al., 2025) enable models to communicate through model internal states such as hidden representations (Coda-Forno et al., 2025) for efficiency (Kong et al., 2026). Most existing methods rely on auxiliary alignment mechanisms and typically require additional training to facilitate effective cross-model communication. On the other hand, LatentMAS is a training-free latent multi-agent system in which each agent reasons and communicates via native latent representations.

We leave additional related works on latent reasoning and agent collaboration in Appendix H.

## 7. Conclusion

We introduce LatentMAS, a training-free framework that enables multi-agent systems to collaborate entirely within latent space. Our framework enables latent auto-regressive reasoning with a lossless latent working-memory transfer mechanism. Theoretical analyses and empirical evaluations across diverse reasoning and generation benchmarks demonstrate that LatentMAS consistently improves accuracy performance, token usage, and decoding speed over strong single-model and text-based MAS baselines. LatentMAS establishes a scalable and general paradigm for agentic collaboration beyond the constraints of natural language.

## Acknowledgment

The authors thank members of the iDEA-iSAIL Lab and the Zou Group for helpful discussions and feedback on this work. This work is supported by National Science Foundation under Award No. IIS-2433308 and Award No. IIS-2117902. The views and conclusions are those of the authors and should not be interpreted as representing the official policies of the funding agencies or the government.

## Impact Statement

This paper introduces LatentMAS, a framework for multi-agent collaboration in the latent space, aiming to improve reasoning performance while reducing inference cost. The methods and findings in this work are intended to advance research in efficient and scalable machine learning systems. We do not anticipate immediate negative societal impacts beyond those commonly associated with deploying more capable and efficient language model systems.

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

# Table of Contents

# Appendix

## A. Input-Output Alignment in LatentMAS

### A.1. Theoretical Justification on $W_a$

In this section, we outline the theoretical justification for how $W_a$ minimizes the distributional gap between the distribution of token embeddings and the distribution of aligned embeddings.

Let $P_e$ and $P_h$ be the distribution of token embeddings $e$ and the hidden embeddings $h$, respectively. We assume that $P_e$ and $P_h$ can be generated by $e = \sum_{x \in \mathcal{V}} \sigma_x W_{\text{in},x}$ and $h = \beta \sum_{x \in \mathcal{V}} \sigma_x W_{\text{out},x}$, respectively, where each semantic $\sigma \in \mathbb{R}^{\mathcal{V}}$ is represented as a weight vector over tokens $\mathcal{V}$ with $\|\sigma\|_2 \leq 1$ and follows an underlying discrete semantic distribution $\sigma \sim P_\sigma$, and $\beta > 0$ characterizes the scale of output embeddings $h$. In practice, we calculate the scale $\beta$ by $\frac{\mathcal{V}\|h\|}{\sum_{x \in \mathcal{V}} \|W_{\text{in},x}\|}$. For an alignment matrix $W_a$, the aligned embedding distribution $P_{\hat{e},W_a}$ is

$$P_{\hat{e},W_a} : \ \hat{e} = hW_a, \quad h \sim P_h. \tag{5}$$

Our goal is to minimize the distance between the aligned embedding distribution $P_{\hat{e},W_a}$ and the token embedding distribution $P_e$, which we measure via the Wasserstein distance:

$$d_{\text{Wasserstein}}(P_{\hat{e},W_a}, P_e) := \inf_{\gamma \in \Gamma(P_e, P_{\hat{e},W_a})} \sqrt{\mathbb{E}_{(\hat{e},e) \sim \gamma}[\|\hat{e} - e\|_2^2]}, \tag{6}$$

where $\Gamma(P_{\hat{e},W_a}, P_e)$ is the set of all couplings of $P_e$ and $P_{\hat{e},W_a}$.

---

**Theorem A.1** (**Upper Bound on Distribution Alignment**). *For any alignment matrix $W_a$, the Wasserstein distance between $P_e$ and $P_{\hat{e},W_a}$ is upper bounded by*

$$d_{\text{Wasserstein}}(P_{\hat{e},W_a}, P_e) \leq \|\beta W_{\text{out}} W_a - W_{\text{in}}\|_F. \tag{7}$$

*As we show in Appendix A.2, our choice of $W_a$ (Equation 3) minimizes this upper bound of $W(P_{\hat{e},W_a}, P_e)$.*

---

*Proof.* Consider the following joint distribution $\gamma^*(\hat{e}, e)$:

$$\gamma^*(\hat{e}, e) := \sum_{\sigma \in \text{supp}(P_\sigma)} P_\sigma(\sigma) 1_{[\beta \sum_{x \in \mathcal{V}} \sigma_x W_{\text{out},x} W_a = \hat{e}]} 1_{[\sum_{x \in \mathcal{V}} \sigma_x W_{\text{in},x} = e]}. \tag{8}$$

Note that for every $\hat{e}$,

$$\sum_{e \in \text{supp}(P_e)} \gamma^*(\hat{e}, e) = \sum_{e \in \text{supp}(P_e)} \sum_{\sigma \in \text{supp}(P_\sigma)} P_\sigma(\sigma) 1_{[\beta \sum_{x \in \mathcal{V}} \sigma_x W_{\text{out},x} W_a = \hat{e}]} 1_{[\sum_{x \in \mathcal{V}} \sigma_x W_{\text{in},x} = e]} \tag{9}$$

$$= \sum_{\sigma \in \text{supp}(P_\sigma)} P_\sigma(\sigma) 1_{[\beta \sum_{x \in \mathcal{V}} \sigma_x W_{\text{out},x} W_a = \hat{e}]} \sum_{e \in \text{supp}(P_e)} 1_{[\sum_{x \in \mathcal{V}} \sigma_x W_{\text{in},x} = e]} \tag{10}$$

$$= \sum_{\sigma \in \text{supp}(P_\sigma)} P_\sigma(\sigma) 1_{[\beta \sum_{x \in \mathcal{V}} \sigma_x W_{\text{out},x} W_a = \hat{e}]} 1_{[\sum_{x \in \mathcal{V}} \sigma_x W_{\text{in},x} \in \text{supp}(P_e)]} \tag{11}$$

$$= \sum_{\sigma \in \text{supp}(P_\sigma)} P_\sigma(\sigma) 1_{[\beta \sum_{x \in \mathcal{V}} \sigma_x W_{\text{out},x} W_a = \hat{e}]} \tag{12}$$

$$= P_{\hat{e},W_a}(\hat{e}); \tag{13}$$

and that for every $e$,

$$\sum_{\hat{e} \in \text{supp}(P_{\hat{e}, W_a})} \gamma^*(\hat{e}, e) = \sum_{\hat{e} \in \text{supp}(P_{\hat{e}, W_a})} \sum_{\sigma \in \text{supp}(P_\sigma)} P_\sigma(\sigma) 1_{[\beta \sum_{x \in \mathcal{V}} \sigma_x W_{\text{out},x} W_a = \hat{e}]} 1_{[\sum_{x \in \mathcal{V}} \sigma_x W_{\text{in},x} = e]} \tag{14}$$

$$= \sum_{\sigma \in \text{supp}(P_\sigma)} P_\sigma(\sigma) 1_{[\sum_{x \in \mathcal{V}} \sigma_x W_{\text{in},x} = e]} \sum_{\hat{e} \in \text{supp}(P_{\hat{e}, W_a})} 1_{[\beta \sum_{x \in \mathcal{V}} \sigma_x W_{\text{out},x} W_a = \hat{e}]} \tag{15}$$

$$= \sum_{\sigma \in \text{supp}(P_\sigma)} P_\sigma(\sigma) 1_{[\sum_{x \in \mathcal{V}} \sigma_x W_{\text{in},x} = e]} 1_{[\beta \sum_{x \in \mathcal{V}} \sigma_x W_{\text{out},x} W_a \in \text{supp}(P_{\hat{e}, W_a})]} \tag{16}$$

$$= \sum_{\sigma \in \text{supp}(P_\sigma)} P_\sigma(\sigma) 1_{[\sum_{x \in \mathcal{V}} \sigma_x W_{\text{in},x} = e]} \tag{17}$$

$$= P_e(e). \tag{18}$$

This implies $\gamma^* \in \Gamma(P_{\hat{e}, W_a}, P_e)$. Since $\sigma$ is a distribution over tokens, and $\| \cdot \|_2^2$ is a convex function. Finally, since $\|\sigma\|_2 \leq$ for all semantic $\sigma \in \text{supp}(P_\sigma)$, we have

$$d_{\text{Wasserstein}}(P_{\hat{e}, W_a}, P_e) = \inf_{\gamma \in \Gamma(P_e, P_{\hat{e}, W_a})} \sqrt{\mathbb{E}_{(\hat{e}, e) \sim \gamma}[\|\hat{e} - e\|_2^2]} \tag{19}$$

$$\leq \sqrt{\mathbb{E}_{(\hat{e}, e) \sim \gamma^*}[\|\hat{e} - e\|_2^2]} \tag{20}$$

$$= \sqrt{\sum_{\hat{e} \in \text{supp}(P_{\hat{e}, W_a})} \sum_{e \in \text{supp}(P_e)} \gamma^*(\hat{e}, e) \|\hat{e} - e\|_2^2} \tag{21}$$

$$= \sqrt{\sum_{\hat{e} \in \text{supp}(P_{\hat{e}, W_a})} \sum_{e \in \text{supp}(P_e)} \sum_{\sigma \in \text{supp}(P_\sigma)} P_\sigma(\sigma) 1_{[\beta \sum_{x \in \mathcal{V}} \sigma_x W_{\text{out},x} W_a = \hat{e}]} 1_{[\sum_{x \in \mathcal{V}} \sigma_x W_{\text{in},x} = e]} \|\hat{e} - e\|_2^2} \tag{22}$$

$$= \sqrt{\sum_{\sigma \in \text{supp}(P_\sigma)} P_\sigma(\sigma) \sum_{\hat{e} \in \text{supp}(P_{\hat{e}, W_a})} \sum_{e \in \text{supp}(P_e)} 1_{[\beta \sum_{x \in \mathcal{V}} \sigma_x W_{\text{out},x} W_a = \hat{e}]} 1_{[\sum_{x \in \mathcal{V}} \sigma_x W_{\text{in},x} = e]} \|\hat{e} - e\|_2^2} \tag{23}$$

$$= \sqrt{\sum_{\sigma \in \text{supp}(P_\sigma)} P_\sigma(\sigma) \left\| \beta \sum_{x \in \mathcal{V}} \sigma_x W_{\text{out},x} W_a - \sum_{x \in \mathcal{V}} \sigma_x W_{\text{in},x} \right\|_2^2} \tag{24}$$

$$= \sqrt{\sum_{\sigma \in \text{supp}(P_\sigma)} P_\sigma(\sigma) \left\| \sum_{x \in \mathcal{V}} \sigma_x (\beta W_{\text{out},x} W_a - W_{\text{in},x}) \right\|_2^2} \tag{25}$$

$$= \sqrt{\sum_{\sigma \in \text{supp}(P_\sigma)} P_\sigma(\sigma) \|\sigma(\beta W_{\text{out}} W_a - W_{\text{in}})\|_2^2} \tag{26}$$

$$\leq \sqrt{\sum_{\sigma \in \text{supp}(P_\sigma)} P_\sigma(\sigma) \|\sigma\|_2^2 \|\beta W_{\text{out}} W_a - W_{\text{in}}\|_2^2} \tag{27}$$

$$\leq \sqrt{\sum_{\sigma \in \text{supp}(P_\sigma)} P_\sigma(\sigma) \|\beta W_{\text{out}} W_a - W_{\text{in}}\|_2^2} \tag{28}$$

$$\leq \sqrt{\sum_{\sigma \in \text{supp}(P_\sigma)} P_\sigma(\sigma) \|\beta W_{\text{out}} W_a - W_{\text{in}}\|_F^2} \tag{29}$$

$$= \sqrt{\|\beta W_{\text{out}} W_a - W_{\text{in}}\|_F^2} \tag{30}$$

$$= \|\beta W_{\text{out}} W_a - W_{\text{in}}\|_F. \tag{31}$$

$\square$

**A.2. Solving the Alignment Matrix $W_a$**

In Section 3.1, we put the last-layer hidden states $h$ back to the input sequence to enable the model's latent reasoning. However, since the $h$ is not perfectly aligned with the input embedding space, directly feeding $h$ into shallow layers may lead to out-of-distribution activation patterns inside LLMs. To mitigate this in a training-free way, we seek a matrix $W_a$ which maps $h$ to a valid input space (i.e., $e = hW_a$). A straightforward way to calculate $W_a$ is to enforce that the aligned latent vector $e$ behaves similarly to a real input embedding when it enters the model. Motivated by our Theorem A.1 below, this corresponds to the following minimization problem:

$$\min_{W_a} \|\beta W_{\text{out}} W_a - W_{\text{in}}\|_F^2. \tag{32}$$

This objective is quadratic in $W_a$, so we can derive a closed-form solution by setting its derivative to zero, which yields the normal equation:

$$\beta W_{\text{out}}^\top W_{\text{out}} W_a - W_{\text{out}}^\top W_{\text{in}} = 0. \tag{33}$$

Solving for $W_a$ gives:

$$W_a = \frac{1}{\beta} \left( W_{\text{out}}^\top W_{\text{out}} \right)^{-1} W_{\text{out}}^\top W_{\text{in}}. \tag{34}$$

For numerical stability, we further add a small hyperparameter $\lambda > 0$ to obtain a ridge regression solution (Hoerl & Kennard, 1970):

$$W_a = \frac{1}{\beta} \left( W_{\text{out}}^\top W_{\text{out}} + \lambda I \right)^{-1} W_{\text{out}}^\top W_{\text{in}}, \tag{35}$$

which we compute once and reuse for all latent reasoning steps. Here, $\lambda > 0$ is a small hyperparameter to approximate the pseudo-inverse $W_{\text{out}}^\dagger = \lim_{\lambda \searrow 0} \left( W_{\text{out}}^\top W_{\text{out}} + \lambda I \right)^{-1} W_{\text{out}}^\top$.

# B. Theoretical Analysis

## B.1. Proof of Theorem 3.1

**Assumption B.1** (Linear Representation Hypothesis; Park et al., 2023b). *We assume that the hidden embeddings $h$ are linear combinations $\sum_{i=1}^{d_h} c_i s_i$ of an underlying semantic basis $\{s_1, \ldots, s_{d_h}\} \subset \mathbb{R}^{d_h}$ (linearly independent) with ternary coefficients $c_1, \ldots, c_{d_h} \in \{0, \pm 1\}$, where $c_i = 0$ represents that $h$ does not have semantic $i$, and $c_i = \pm 1$ represents that $h$ has semantic $i$ in a positive/negative way.*

---

**Theorem B.1** (**Restate of Theorem 3.1**). *Under the Linear Representation Hypothesis on $h$, if the sequence of all latent thoughts with length $m$ can be expressed losslessly through corresponding text-based reasoning, then the length of text (in tokens) needs to be at least $\Omega(d_h m / \log |\mathcal{V}|)$, where $|\mathcal{V}| > 1$ denotes the vocabulary size.*

---

*Proof of Theorem 3.1.* Under Assumption B.1, the set $\mathcal{H}$ of hidden embeddings is

$$\mathcal{H} = \left\{ \sum_{i=1}^{d_h} c_i s_i : c_1, \ldots, c_{d_h} \in \{0, \pm 1\} \right\}, \tag{36}$$

where $\{s_1, \ldots, s_{d_h}\} \subset \mathbb{R}^{d_h}$ is the underlying semantic basis. Then, the set of length-$t$ latent reasoning sequences is $\mathcal{H}^m$. Since the semantic basis is linearly independent, the size of the set $\mathcal{H}$ of hidden embeddings is

$$|\mathcal{H}| = |\{0, \pm 1\}|^{|\{s_1, \ldots, s_{d_h}\}|} = 3^{d_h}. \tag{37}$$

Thus, the size of the set of length-$m$ latent reasoning sequences is

$$|\mathcal{H}^m| = |\mathcal{H}|^m = (3^{d_h})^m = 3^{d_h m}. \tag{38}$$

To represent the set $\mathcal{H}^m$ of length-$m$ latent reasoning sequences via the set $\mathcal{V}^{m'}$ of length-$m'$ text-based reasoning sequences losslessly, there needs to exist an surjective map from $\mathcal{V}^{m'}$ to $\mathcal{H}^m$, which implies that $|\mathcal{V}^{m'}| \geq |\mathcal{H}^m|$. Therefore,

$$m' = \log_{|\mathcal{V}|}(|\mathcal{V}|^{m'}) = \log_{|\mathcal{V}|} |\mathcal{V}^{m'}| \tag{39}$$

$$\geq \log_{|\mathcal{V}|} |\mathcal{H}^m| = \log_{|\mathcal{V}|}(3^{d_h m}) \tag{40}$$

$$= \frac{d_h m \log 3}{\log |\mathcal{V}|} = \Omega\left(\frac{d_h m}{\log |\mathcal{V}|}\right). \qquad \square$$

## B.2. Proof of Theorem 3.3

---

**Theorem B.2** (**Restate of Theorem 3.3**). *In both latent and text-based reasoning, the outputs of an agent when receiving latent working memory from preceding agents are equivalent to those obtained when directly inputting the preceding agents' outputs.*

---

*Proof.* Let $h^{(l)}, K^{(l)}, V^{(l)}$ and $h'^{(l)}, K'^{(l)}, V'^{(l)}$ denote the output, keys, and values of $l$-th transformer layer when receiving latent working memory from preceding agents and when directly inputting the preceding agents' outputs, respectively. In the following, we will use induction to show that $h^{(l)} = h'^{(l)}$ for every layer $l = 1, \ldots, L$.

**Induction step.** Suppose that $h^{(l-1)} = h'^{(l-1)}$, and we will show that $h^{(l)} = h'^{(l)}$.

The KV cache contains $K_{\leq t+m}{}^{(l)}$ and $V_{\leq t+m}{}^{(l)}$. For each past token layers, at each attention layer, the transformer produces one column of $K_{\leq t+m}{}^{(l)}$ and a corresponding column of $V_{\leq t+m}{}^{(l)}$. At the next step the model forms a query from the current input and then uses that query together with the stored $K_{\leq t+m}{}^{(l)}$ and $V_{\leq t+m}{}^{(l)}$ to form the attention result. That attention result is a deterministic function of the query and of the keys and values it attends to.

We are comparing two ways to make those same keys and values available to the current computation: (i) actually feeding the earlier tokens into the model again, in which case the model will recompute the same keys and values and then use them in attention; (ii) reading in $K_{\leq t+m}{}^{(l)}$ and $V_{\leq t+m}{}^{(l)}$ from the cache and use them directly. In both cases the keys and values presented to the attention computation are identical, because the cache was produced by the same model on the same inputs.

Given identical keys and values and the same current input, the attention output is the same in both scenarios. The remainder of the transformer computation that produces the last-layer hidden embedding is a deterministic function of that attention output (and the current input). Therefore the last-layer hidden embedding $h^{(l)}$ produced for the current step is the same whether the model recomputed keys/values from tokens or read $K_{\leq t+m}^{(l)}, V_{\leq t+m}^{(l)}$ from cache. Formally, since $h^{(l-1)} = h'^{(l-1)}$, $K_{\leq t+m}^{(l)} = K'_{\leq t+m}^{(l)}$, and $V_{\leq t+m}^{(l)} = V'_{\leq t+m}^{(l)}$, then $h^{(l)} = h'^{(l)}$.

**Induction base case.** For the first layer, similarly with the induction step, since the input is the same (for both latent-based and text-based reasoning), $K_{\leq t+m}^{(1)} = K'_{\leq t+m}^{(1)}$, and $V_{\leq t+m}^{(1)} = V'_{\leq t+m}^{(1)}$, then $h^{(1)} = h'^{(1)}$.

**Conclusion.** By induction, we have that $h^{(l)} = h'^{(l)}$ of every layer $l = 1, \ldots, L$. In particular, since $h = h^{(L)}$ and $h' = h'^{(L)}$, then $h = h^{(L)} = h'^{(L)} = h'$. $\qquad\square$

## B.3. Proof of Theorem 3.4

> **Theorem B.3** (**Restate of Theorem 3.4**). *The time complexity for each agent of LatentMAS is $O\big((d_h^2 m + d_h m^2 + d_h tm)L\big)$, where $t$ is the input length of this agent, and $m$ is the length of latent thoughts. In contrast, assuming Theorem 3.1, the time complexity for each agent of the vanilla text-based MAS needs to be $O\big((d_h^3 m \frac{1}{\log|\mathcal{V}|} + d_h^3 m^2 \frac{1}{\log^2|\mathcal{V}|} + d_h^2 tm \frac{1}{\log|\mathcal{V}|})L + d_h^2|\mathcal{V}|m \frac{1}{\log|\mathcal{V}|}\big)$ to achieve the same expressiveness.*

*Proof.* We analyze the time complexity of our LatentMAS and the vanilla text-based MAS separately.

**Time complexity of our method.** Recall that a transformer layer consists of two main components: self-attention and feed-forward networks. For a length-$(t+m)$ sequence, the time complexity to compute self-attention for $m$ latent reasoning steps is $O(d_h(t+m)m) = O(d_h(m^2 + tm))$ due to the attention computation between $O(t^2)$ pairs of tokens, and the time complexity to compute feed-forward networks for $m$ latent reasoning steps is $O(d_h^2 m)$ due to matrix–vector multiplication. Since there are $L$ layers, the overall time complexity of our method is

$$O\big((d_h(m^2 + tm) + d_h^2 m)L\big). \tag{41}$$

**Time complexity of the vanilla text-based MAS.** Let $m'$ denote the number of text-based reasoning steps. Similarly with the complexity analysis of our method, the time complexity to compute the hidden embeddings is

$$O\big((d_h(m'^2 + tm') + d_h^2 m')L\big). \tag{42}$$

Besides that, due to matrix–vector multiplication and softmax computation, the time complexity to decode hidden embeddings into tokens is

$$O\big(d_h|\mathcal{V}|m'\big). \tag{43}$$

Hence, the overall time complexity of the vanilla MAS is

$$O\big((d_h(m'^2 + tm') + d_h^2 m')L + d_h|\mathcal{V}|m'\big). \tag{44}$$

Assuming Theorem 3.1, the number of text-based reasoning steps is

$$m' = O\Big(\frac{d_h m}{\log|\mathcal{V}|}\Big). \tag{45}$$

It follows that the overall time complexity is

$$O\big((d_h(m'^2 + tm) + d_h^2 m')L + d_h|\mathcal{V}|m'\big) \tag{46}$$

$$= \Big(\Big(d_h\Big(\Big(\frac{d_h m}{\log|\mathcal{V}|}\Big)^2 + t\Big(\frac{d_h m}{\log|\mathcal{V}|}\Big)\Big) + d_h^2\Big(\frac{d_h m}{\log|\mathcal{V}|}\Big)\Big)L + d_h|\mathcal{V}|\Big(\frac{d_h m}{\log|\mathcal{V}|}\Big)\Big) \tag{47}$$

$$= O\Big(\Big(\frac{d_h^3 m^2}{\log^2|\mathcal{V}|} + \frac{d_h^3 m}{\log|\mathcal{V}|} + \frac{d_h^2 tm}{\log|\mathcal{V}|}\Big)L + \frac{d_h^2|\mathcal{V}|m}{\log|\mathcal{V}|}\Big). \tag{48}$$

$$\square$$

# C. Experiment Setups

## C.1. Evaluation Details

We introduce all datasets used in our experiments as follows:

**Math & Science Reasoning.**

- **GSM8K** (Cobbe et al., 2021) is a widely used benchmark of 8.5K grade-school math word problems designed to evaluate multi-step numerical reasoning. Each problem requires decomposing a natural-language description into structured arithmetic steps, making it a standard testbed for assessing chain-of-thought reasoning ability.

- **AIME24** (Maxwell-Jia, 2024) consists of 30 competition-level problems from the 2024 American Invitational Mathematics Examination. These questions span algebra, geometry, number theory, and combinatorics, and require precise numeric answers with typically 1–3 digits, making the benchmark a compact but challenging evaluation of high-school Olympiad-style reasoning.

- **AIME25** (math ai, 2025) provides 30 additional problems from the 2025 AIME exam, maintaining the same answer format and difficulty profile. Compared with AIME24, this benchmark includes more multi-phase derivations and intricate combinatorial constructions, offering a complementary stress test for mathematical robustness.

- **GPQA-Diamond** (Rein et al., 2023) is the most difficult split of the GPQA benchmark with 198 questions, featuring graduate-level multiple-choice questions written by domain experts in physics, biology, and chemistry. The dataset emphasizes conceptual depth, cross-disciplinary reasoning, and the ability to synthesize multi-step scientific arguments under rigorous distractor settings.

- **MedQA** (Yang et al., 2024a) contains real medical licensing exam questions that assess biomedical knowledge, clinical reasoning, and diagnostic decision-making. Problems require integrating textual context with domain-specific medical understanding, making the benchmark a representative testbed for professional-level scientific reasoning.

**Commonsense Reasoning.**

- **ARC-Easy** (Clark et al., 2018b) consists of grade-school science questions from the AI2 Reasoning Challenge that test foundational factual knowledge and straightforward commonsense reasoning. As a simplified subset of ARC, it serves as a baseline measure of basic scientific understanding without requiring complex multi-step inference.

- **ARC-Challenge** (Clark et al., 2018a) includes the most difficult items from the AI2 Reasoning Challenge. These questions are intentionally adversarial, requiring multi-hop reasoning, causal and counterfactual inference, and systematic elimination of distractor choices. Performance on ARC-Challenge is widely regarded as a strong indicator of robust commonsense reasoning capabilities.

**Code Generation.**

- **MBPP-Plus** (Liu et al., 2023) extends the original MBPP benchmark with broader input coverage, additional hidden test cases, and stricter execution-based evaluation. Each problem requires generating a self-contained Python function that satisfies a comprehensive unit-test suite, making the benchmark a robust measure of code synthesis reliability and correctness.

- **HumanEval-Plus** (Liu et al., 2023) augments HumanEval with denser, more challenging test suites, significantly increasing the rigor of functional correctness evaluation. The benchmark emphasizes generalization beyond prompt examples and tests a model's ability to produce semantically precise, executable Python code under more demanding verification settings.

## C.2. Implementation Details

In addition to the experimental setups described in the main paper, we provide additional implementation and evaluation details below.

**Software Backend**   All methods are implemented in Python using PyTorch and HuggingFace `Transformers`, with an optional `vLLM` backend for fast decoding and tensor-parallel inference. We use the official chat templates and special tokens such as `<|im_start|>` and `<|im_end|>`.

**Evaluation protocol.**   For all non-coding benchmarks, we report accuracy based on answer matching of the final answer after text normalization (lowercasing, trimming whitespace, and removing extraneous punctuation).

For multiple-choice datasets (GPQA-Diamond, MedQA, ARC-Easy, ARC-Challenge), we first extract the model's final answer string and then compare it via exact match to the answer letter.

For numeric problems (GSM8k, AIME24, AIME25), we evaluate correctness based on numeric equality: we extract the final predicted answer, parse both prediction and answer into numbers, and mark as correct only if the two values match. Predictions that fail numeric parsing are counted as incorrect.

For code generation tasks (MBPP-Plus and HumanEval-Plus), we evaluate the code by executing unit tests. Specifically, we extract the predicted code from model's output, append the ground-truth tests provided by the benchmark, and execute the combined script in a sandboxed environment with a 10-second timeout. A sample is counted as correct if and only if all tests pass without runtime errors.

## D. Additional Experiments

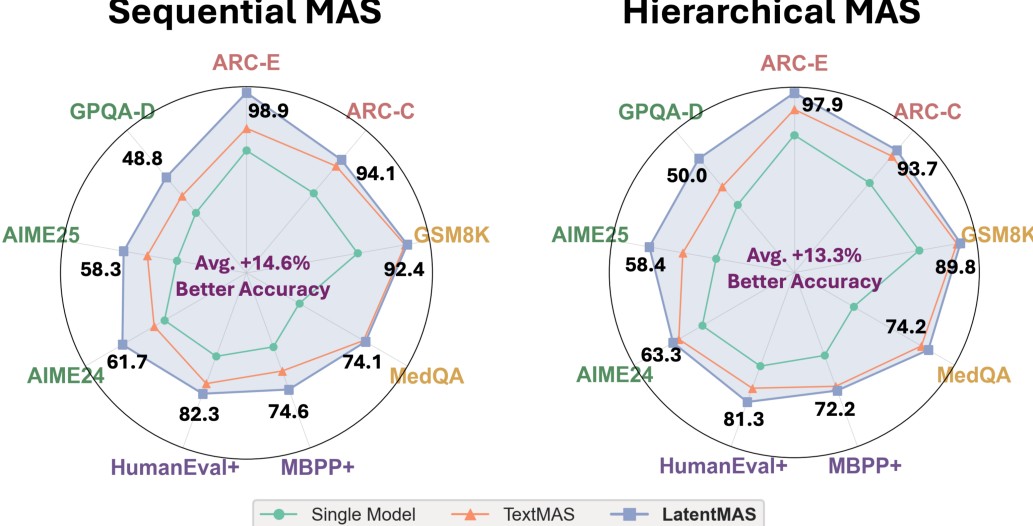

*Figure 9.* Illustration of the accuracy gains via **LatentMAS** across the Sequential and Hierarchical MAS settings. LatentMAS consistently improves system-level reasoning accuracy compared with single model and text-based MAS.

### D.1. Performance of LatentMAS under Hierarchical MAS Setting

*Table 3.* **Main results of LatentMAS on 6 general tasks under the Hierarchical MAS setting.** We report accuracy, token usage, and end-to-end speed, and highlight the performance gains following the same evaluation protocol as in Table 1.

| Tasks | Metrics | Qwen3-4B | | | Improve | Qwen3-8B | | | Improve | Qwen3-14B | | | Improve |
|---|---|---|---|---|---|---|---|---|---|---|---|---|---|
| | | Single | TextMAS | LatentMAS | | Single | TextMAS | LatentMAS | | Single | TextMAS | LatentMAS | |
| *Hierarchical MAS Setting* | | | | | | | | | | | | | |
| **ARC-E** | Acc. | 95.4 | 97.1 | 96.8 | ↓0.3 | 95.6 | 98.2 | **98.3** | ↑0.1 | 97.2 | 98.3 | **98.7** | ↑0.4 |
| | Token | 724 | 2054 | **363** | ↓82.3% | 656 | 2237 | **308** | ↓86.2% | 608 | 2752 | **619** | ↓77.5% |
| | Speed | 369 | 2239 | **591** | ×3.8 | 404 | 3619 | **1779** | ×2.0 | 551 | 7102 | **1884** | ×3.8 |
| **ARC-C** | Acc. | 89.2 | 92.5 | 91.7 | ↓0.8 | 91.0 | 93.3 | **93.9** | ↑0.6 | 92.6 | 95.3 | **95.5** | ↑0.2 |
| | Token | 913 | 2674 | **447** | ↓83.3% | 846 | 2854 | **344** | ↓87.9% | 773 | 2167 | **295** | ↓86.4% |
| | Speed | 97 | 1275 | **299** | ×4.3 | 266 | 2034 | **714** | ×2.8 | 338 | 4283 | **1090** | ×3.9 |
| **GSM8K** | Acc. | 82.4 | 89.4 | 88.4 | ↓1.0 | 81.1 | 90.4 | 89.5 | ↓0.9 | 83.7 | 90.8 | **91.6** | ↑0.8 |
| | Token | 1136 | 3098 | **555** | ↓82.1% | 1280 | 2370 | **353** | ↓85.1% | 1118 | 3021 | **495** | ↓83.6% |
| | Speed | 469 | 1878 | **360** | ×5.2 | 449 | 1365 | **702** | ×1.9 | 536 | 3675 | **1631** | ×2.3 |
| **MedQA** | Acc. | 47.7 | 65.0 | **67.3** | ↑2.3 | 53.0 | 76.3 | **77.0** | ↑0.7 | 64.7 | 78.0 | **78.3** | ↑0.3 |
| | Token | 2134 | 6702 | **1015** | ↓84.9% | 2098 | 6893 | **1007** | ↓85.4% | 1746 | 5473 | **899** | ↓83.6% |
| | Speed | 236 | 1495 | **557** | ×2.7 | 476 | 3387 | **964** | ×3.5 | 1360 | 7591 | **1250** | ×6.1 |
| **MBPP+** | Acc. | 63.5 | 69.3 | **70.6** | ↑1.3 | 64.8 | 71.9 | **72.2** | ↑0.3 | 68.5 | 73.0 | **73.8** | ↑0.8 |
| | Token | 1634 | 6782 | **1339** | ↓80.3% | 2053 | 7703 | **1264** | ↓83.6% | 1858 | 7458 | **1187** | ↓84.1% |
| | Speed | 523 | 1766 | **489** | ×3.6 | 1064 | 3898 | **1387** | ×2.8 | 2410 | 9162 | **2507** | ×3.7 |
| **HumanEval+** | Acc. | 75.0 | 76.2 | **79.3** | ↑3.1 | 74.4 | 76.8 | **78.0** | ↑1.2 | 76.8 | 84.1 | **86.6** | ↑2.5 |
| | Token | 2380 | 8127 | **1373** | ↓83.1% | 2507 | 8768 | **1274** | ↓85.5% | 2366 | 8114 | **1512** | ↓81.4% |
| | Speed | 274 | 931 | **333** | ×2.8 | 502 | 1809 | **439** | ×4.1 | 1084 | 3988 | **1188** | ×3.4 |

We report additional results on three Qwen3 backbones under the hierarchical MAS setting to complement the main experiments. As shown in Table 3, LatentMAS consistently achieves higher or comparable system-level accuracy relative to TextMAS while substantially reducing both end-to-end inference time and total system token usage. These results mirror the trends observed in the sequential setting and confirm that latent working-memory transfer remains effective in hierarchical collaboration, where multiple expert agents contribute to the final prediction.

## D.2. LatentMAS on Llama Backbones

*Table 4.* **Main results of LatentMAS on Llama3 families under Sequential and Hierarchical MAS settings.** We report accuracy, token usage, and end-to-end speed, and highlight the performance gains following the same evaluation protocol as in Table 1.

| Tasks | Metrics | Llama-3.2-3B-Instruct | | | Improve | Llama-3.1-8B-Instruct | | | Improve |
|---|---|---|---|---|---|---|---|---|---|
| | | Single | TextMAS | LatentMAS | | Single | TextMAS | LatentMAS | |
| *Sequential MAS Setting* | | | | | | | | | |
| **ARC-E** | Acc. | 79.2 | 80.1 | **80.4** | ↑ **0.3** | 85.7 | 89.6 | **91.2** | ↑ **1.6** |
| | Token | 236 | 1889 | **229** | ↓ **87.9%** | 219 | 1981 | **228** | ↓ **88.5%** |
| | Speed | 155 | 1670 | **392** | ×**4.3** | 352 | 2708 | **516** | ×**5.2** |
| **MedQA** | Acc. | 46.7 | 48.0 | 48.0 | ↑ 0.0 | 59.0 | 65.0 | **65.3** | ↑ **0.3** |
| | Token | 405 | 1595 | **403** | ↓ **74.7%** | 382 | 2354 | **316** | ↓ **86.6%** |
| | Speed | 51 | 236 | **85** | ×**2.8** | 112 | 589 | **182** | ×**3.2** |
| **HumanEval+** | Acc. | 39.6 | 43.2 | **44.5** | ↑ **1.3** | 51.2 | 55.4 | **56.7** | ↑ **1.3** |
| | Token | 227 | 1042 | **338** | ↓ **67.6%** | 161 | 740 | **218** | ↓ **70.5%** |
| | Speed | 64 | 199 | **77** | ×**2.6** | 80.6 | 333 | **101** | ×**3.3** |
| *Hierarchical MAS Setting* | | | | | | | | | |
| **ARC-E** | Acc. | 79.2 | 81.6 | **82.3** | ↑ **0.7** | 85.7 | 89.3 | **90.4** | ↑ **1.1** |
| | Token | 236 | 956 | **177** | ↓ **81.5%** | 219 | 1118 | **206** | ↓ **81.6%** |
| | Speed | 155 | 615 | **192** | ×**3.2** | 352 | 1832 | **669** | ×**2.7** |
| **MedQA** | Acc. | 46.7 | 50.3 | 50.0 | ↓ 0.3 | 59.0 | 64.3 | **67.7** | ↑ **3.4** |
| | Token | 405 | 1729 | **359** | ↓ **79.2%** | 382 | 1847 | **191** | ↓ **89.7%** |
| | Speed | 51 | 282 | **73** | ×**3.9** | 112 | 464 | **204** | ×**2.3** |
| **HumanEval+** | Acc. | 39.6 | 43.9 | **45.7** | ↑ **1.8** | 51.2 | 54.2 | **55.4** | ↑ **1.2** |
| | Token | 227 | 719 | **226** | ↓ **68.6%** | 161 | 414 | **187** | ↓ **54.8%** |
| | Speed | 64 | 133 | **72** | ×**1.8** | 80.6 | 156 | **88** | ×**1.8** |

We report additional sequential and hierarchical MAS results on Llama-3 backbones to complement the main evaluations. Consistent with Table 1 and 2 in the main paper, LatentMAS maintains higher or comparable accuracy relative to TextMAS while consistently reducing system-wise token usage and inference time. Across tasks, LatentMAS preserves the same qualitative trend observed for Qwen models, achieving faster end-to-end inference with markedly fewer decoded tokens, indicating that LatentMAS exhibits strong generalizability across different model families.

## D.3. Additional Efficiency Analyses on LatentMAS

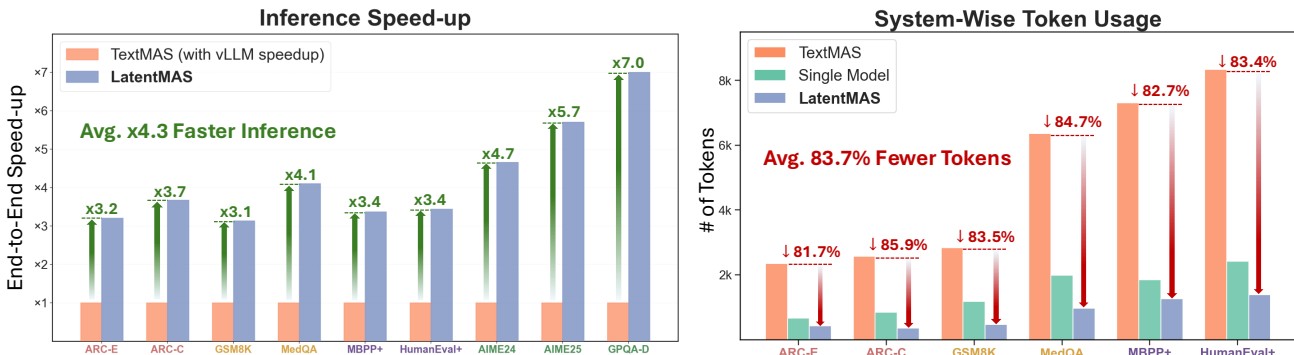

*Figure 10.* Efficiency gains of LatentMAS over single model and TextMAS under the Hierarchical MAS setting. **Left:** LatentMAS achieves consistent faster end-to-end inference speed. **Right:** LatentMAS requires fewer system-wise token usage in total.

*Table 5.* Token usage of the final agent in MAS. LatentMAS consistently reduces the final decoding length compared with TextMAS.

| Method | Size | ARC-C | GSM8K | MedQA | HumanEval+ | GPQA-D |
|--------|------|-------|-------|-------|------------|--------|
| TextMAS | 4B | 614 | 607 | 1539 | 2037 | – |
| LatentMAS | 4B | **447** | **555** | **1015** | **1373** | – |
| TextMAS | 8B | 685 | 412 | 1664 | 2167 | 4758 |
| LatentMAS | 8B | **344** | **353** | **1007** | **1274** | **3395** |
| TextMAS | 14B | 492 | 573 | 1303 | 2008 | 4517 |
| LatentMAS | 14B | **295** | **495** | **899** | **1512** | **3606** |

**Final-Agent Token Usage in LatentMAS.** To further disentangle the source of token savings, we additionally measure the output length of the final answering agent only. This analysis excludes intermediate agents and therefore directly compares the final decoding stage of LatentMAS and TextMAS. As shown in Table 5, the final agent in LatentMAS consistently generates fewer text tokens than the final agent in TextMAS across different model scales and tasks, with an average reduction of 29.1%. This indicates that the token savings in LatentMAS come not only from eliminating intermediate textual communication among agents, but also from shortening the final answer decoding process. This reduction occurs since the final agent in TextMAS receives long textual reasoning traces from preceding agents and often continues the text-based reasoning process before producing the answer. In contrast, the final agent in LatentMAS receives preceding agents' information through compact latent working memory and aggregates these latent thoughts before decoding the final response. As a result, LatentMAS reduces both inter-agent communication tokens and the final-stage decoding length.

# E. Detailed Analyses on LatentMAS

## E.1. Additional Analyses on Input-Output Alignment

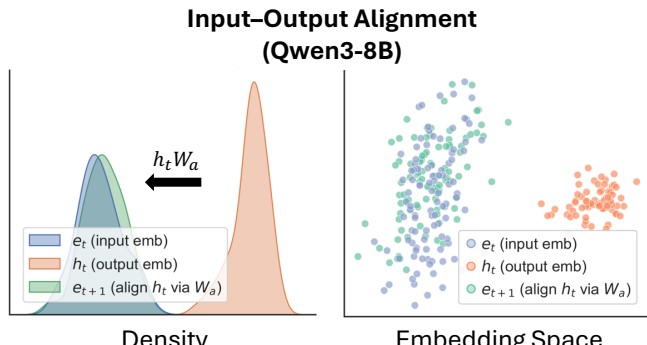

*Figure 11.* **Effectiveness of the input-output alignment $W_a$ on 8B models.** Unaligned output embeddings ($h_t$) drift away from the original input embeddings ($e_t$), while the aligned vectors ($e_{t+1}$) realign with $e_t$, demonstrating that $W_a$ preserves embedding-space structure and prevents representation drift.

In Figure 11, we extend the input-output alignment analysis to additional 8B model scales beyond the Qwen3-4B and Qwen3-14B examples shown in Figure 6. For Qwen3-8B models, the same pattern holds: unaligned latent outputs drift away from the input embedding distribution, while applying the alignment matrix restores geometric and density-level consistency with the original embedding space.

## E.2. Quantifying Latent Thought Diversity.

While Figure 5 provides a qualitative visualization of the embedding distributions, we further quantify the representational diversity of latent thoughts using a cosine-similarity-based metric. Specifically, for each method, we compute the average pairwise cosine similarity among the generated embeddings. A lower value indicates that the representations are less collapsed and more dispersed in the embedding space, corresponding to higher representational diversity.

*Table 6.* Diversity analysis of generated representations. We report the average pairwise cosine similarity among generated embeddings, where lower values indicate less collapsed and more diverse representations.

| Method | Qwen3-4B | Qwen3-8B | Qwen3-14B |
|---|---|---|---|
| TextMAS | 0.126 | 0.142 | 0.155 |
| LatentMAS | **0.104** | **0.093** | **0.108** |

As shown in Table 6, LatentMAS consistently achieves lower average cosine similarity than TextMAS across all three Qwen3 backbones. This confirms that the latent thoughts are not merely aligned with the semantic region of text-based reasoning but also span a broader and less concentrated representation space. Together with Figure 5, these results provide both qualitative and quantitative evidence that latent thoughts preserve semantic consistency with text reasoning while offering richer representational diversity.

## E.3. Ablations on Latent Reasoning and Communication in LatentMAS

To isolate the contribution of each component in LatentMAS, we further conduct hybrid ablations by selectively replacing either latent reasoning or latent communication with its text-based counterpart. Specifically, we compare LatentMAS with the following two variants:

- *Latent Reasoning + Text Communication*, where each agent performs latent reasoning but communicates to the next agent through decoded text, using only the last 128 decoded tokens;

- *Text Reasoning + Latent Communication*, where each agent performs standard text-based reasoning while transferring intermediate information to the next agent through latent working memory.

As shown in Table 7, both hybrid variants underperform full LatentMAS across all evaluated tasks. This demonstrates that the gains of LatentMAS cannot be attributed to only one component. Latent reasoning improves the expressiveness of each agent's internal computation, while latent communication preserves and transfers intermediate information without forcing it through a compressed text interface. Combining both components enables end-to-end latent collaboration and yields the strongest performance.

*Table 7.* Hybrid ablations on latent reasoning and latent communication using Qwen3-8B. Replacing either component with its text-based counterpart leads to consistent performance drops, indicating that both components contribute to the overall gains of LatentMAS.

| Method | GSM8K | MBPP+ | MedQA |
|---|---|---|---|
| Latent Reasoning + Text Communication | 85.5 | 66.4 | 65.9 |
| Text Reasoning + Latent Communication | 90.1 | 68.0 | 71.2 |
| **LatentMAS** | **93.8** | **74.6** | **75.3** |

## F. How to Debug LatentMAS?

In LatentMAS *"debug mode"*, we introduce a switch that enables us to inspect the content of the latent thoughts of agents without altering the overall LatentMAS pipeline. Specifically, when we switch on the *"debug mode"*, each agent should simultaneously generate its latent thoughts and another text response within the same contextual information (e.g., agent prompt and latent working memory). The latent thoughts are passed to the next agent, and the parallel text response serves as a probe for the latent thoughts.

By checking the details of text response of intermediate agents, we can identify the root causes of errors in the final answer. To illustrate this, we present two distinct case studies generated in *"debug mode"* using Qwen3-14B model on GSM8K dataset, corresponding to a failure case and a success case below.

- **Case 1: Incorrect Intermediate Text Leads to Error in LatentMAS Answer**. In the first case of Section I, we examine a problem where LatentMAS produces an incorrect final answer. According to the intermediate agent output, we find the Refiner agent gives an incorrect numerical relationship (as highlighted in red), leading the final Solver agent to utilize this erroneous dependency in its calculation.

- **Case 2: Correct Intermediate Text Leads to Correct LatentMAS Answer**. Conversely, in the second case of Section I, all intermediate agents produce logically sound intermediate outputs, which successfully guides the final agent to derive the correct answer.

Finally, to quantitatively validate that the text in *"debug mode"* serves as a faithful probe for LatentMAS, we conduct a correlation study on 100 debug text - final answer pairs using Qwen3-14B model on GSM8K dataset, which consists of 80 instances with correct final answers and 20 instances with incorrect final answers. We manually annotated the correctness of the each debug text and analyzed their correlation with the final correctness of LatentMAS.

*Table 8.* Correlation analysis between the correctness of intermediate text from *"debug mode"* and the final answers of LatentMAS. The reported statistics demonstrate a high consistency between intermediate text correctness and final answer accuracy.

| | | LatentMAS Final Answer | |
|---|---|---|---|
| | | Correct ($N = 80$) | Incorrect ($N = 20$) |
| Intermediate Debug Text | Correct | **96.2%** (77/80) | 10.0% (2/20) |
| | Incorrect | 3.8% (3/80) | **90.0%** (18/20) |

As shown in Table 8, we observe a high consistency between the correctness of the intermediate text and the final answer. We find that 96.2% of the intermediate text are valid when the final answer is correct, while 90.0% contain errors when the final answer contains error. The result shows that the correctness of intermediate text output aligns well with LatentMAS, validating the *"debug mode"* as a reliable probe for our framework.

## G. Additional Discussions on LatentMAS

**Extension to Heterogeneous Agents.** For simplicity and training-free purposes, we assume that all agents in LatentMAS share the same shape of transformer layers. To relax this assumption and support heterogeneous agents in practice, one can directly leverage prior studies on layer mapping and ensemble learning (Ainsworth et al., 2022; Wortsman et al., 2022) by introducing a trainable adapter to align and share latent representations across different models.

In addition, a potential future direction is to adapt advanced post-training paradigms from text-based MAS to optimize LatentMAS 's latent collaboration protocols to unlock more effective multi-agent reasoning strategies.

## H. Additional Related Works

**Latent Reasoning in LLMs.** Beyond explicit chain-of-thought (CoT) reasoning, recent work has explored the continuous latent space of LLMs as an alternative reasoning medium (Hao et al., 2024; Chen et al., 2025b; Zhu et al., 2025), revealing that hidden states encode richer semantic structures than what discrete token generation can express (Zhang et al., 2025; Liu et al., 2024). Latent reasoning methods such as CoCoNut (Hao et al., 2024) and latent-space editing approaches (e.g., RepE (Zou et al., 2023), LoT (Fungwacharakorn et al., 2024)) demonstrate that manipulating internal representations can guide models to reason more coherently and improve controllability without explicit token-level rationales. Other works (Li et al., 2025a; Wang et al., 2025b; Shi et al., 2025) have also extended latent reasoning paradigms to vision-language models. These methods leverage the structure of hidden states to perform interventions, such as steering, editing, or optimizing latent trajectories, that shape downstream reasoning behavior while remaining agnostic to surface-level text. By operating directly in the continuous space, they can induce reasoning steps that would be difficult or inefficient to express (Zhang et al., 2025; Liu et al., 2024; Coda-Forno et al., 2025). Despite these benefits, existing techniques are confined to a single model's internal computations and do not consider interaction or coordination across multiple reasoning entities (Hao et al., 2024). On the other hand, LatentMAS extends latent reasoning to a multi-agent setting, enabling each agent to generate latent thoughts and propagate latent information to others. Our new framework shifts latent reasoning from an isolated capability of individual models to a system-level collaborative mechanism.

**Agentic Model Collaboration.** With increasing algorithmic advances, LLM-based MAS have been applied across diverse domains (Sagi & Rokach, 2018; Chen et al., 2025a; Yan et al., 2025), such as math and science reasoning (Pezeshkpour et al., 2024; Yue et al., 2024; Wang et al., 2025a), open-domain question answering (Fourney et al., 2024; Wu et al., 2025), and multi-modal GUI interaction (Zhang et al., 2024a; Ye et al., 2025b), demonstrating their versatility in complex real-world settings. A large amount of prior works follow sequential planner-solver pipelines or hierarchical expert-summarizer structures, which correspond to the two MAS settings we adopt for evaluating LatentMAS.

# I. Case Study on Intermediate Error Tracking of LatentMAS

---

**Incorrect Case with LatentMAS (Debug Mode)**

**Question:**
In a room, there are various toys: 5 red cars, 3 action figures, and a doll. The doll costs as much as 3 action figures and one red car costs 4 dollars. How much are all toys worth, of an action figure costs 5 dollars?

---

**Planner:**
Okay, I need to make a detailed plan to solve the math problem.

1. Determine the cost of one action figure (given as $5).

2. Calculate the total cost of the doll; calculate the total cost of all red cars using the given cost of one red car ($4); calculate the total cost of all action figures by multiplying the number of action figures by the cost of one action figure.

3. Add the costs of all toys (red cars, action figures, and the doll) to find the total value of all toys.

**Critic:**
The original plan from the Planner Agent was:

1. Identify the cost of action figure.

2. Calculate the total cost for each type by multiplying the quantity by the individual cost.

3. Sum up all the total costs to find the total value of all toys.

But the problem is that the Planner Agent's plan didn't account for the doll's cost being dependent on the action figures and the red car. So, the original plan might have missed that step. Let me check the original plan again.

But the doll's cost isn't given directly. It's given in terms of other toys. So, the first step should be to calculate the doll's cost based on the given relationship. The original plan might have missed that step, assuming that the cost of each type is given directly, which is not the case for the doll. Therefore, the plan needs to be adjusted to first calculate the doll's cost using the given relationship before proceeding to calculate the total cost for each type.

So, the feedback should point out that the plan didn't account for the doll's cost calculation based on the given relationship. The user needs to first determine the doll's cost by using the values of the action figures and the red car, then proceed with the rest of the steps.

**Refiner:**
The Original Plan is:

1. Identify the cost of each type of toy.

2. Calculate the total cost for each type by multiplying the quantity by the individual cost.

3. Sum up all the total costs to find the total value of all toys.

Feedback: The original plan fails to explicitly address the doll's cost, which is not directly given but derived from the cost of 3 action figures and 1 red car. To refine the plan:

- First, calculate the doll's cost using the given relationship ($3 \times$ action figure cost $+ 1 \times$ red car cost).

- Ensure all quantities (5 red cars, 3 action figures, 1 doll) are accounted for.

- Clarify that the total value includes all toys with their respective quantities and derived costs.

The Refined Plan is:

1. Determine individual costs.

2. Calculate total cost for each toy type.

3. Sum all totals.

**Solver:**
To determine the total value of all the toys, the doll costs as much as 3 action figures and 1 red car, so Doll $= 3 \cdot \$5 + 1 \cdot \$4 = \$19$. Then Red cars $= 5 \cdot \$4 = \$20$, Action figures $= 3 \cdot \$5 = \$15$, Doll $= 1 \cdot \$19 = \$19$. Thus the total value is $\$20 + \$15 + \$19 = \$54$. The final answer is $\boxed{54}$.

---

---

### Correct Case with LatentMAS (Debug Mode)

**Question:**
Jean is two years older than Mark. Two years ago Mark was 5 years older than half Jan's age. If Jan is 30 how old is Jean?

---

**Planner:**

1. Determine Mark's current age by using the information about his age two years ago relative to Jan's age.

2. Calculate half of Jan's age two years ago and then determine Mark's age at that time based on the given relationship.

3. Use the relationship between Jean and Mark's ages to find Jean's current age.

**Critic:**
The user wants to know Jean's age based on the given information. The original plan provided by the Planner Agent was a bit incomplete, so I need to figure out where it went wrong.

The original plan is a good start but lacks specific details on how to calculate Mark's age two years ago and how to derive Jean's age from Mark's current age. To improve, the plan should explicitly outline the mathematical operations needed, such as calculating half of Jan's current age, adding 5 years, adjusting for the two-year time difference, and then applying the age difference between Jean and Mark. Additionally, the plan should emphasize verifying each step for consistency with the given information.

**Refiner:**
First, I need to break down the information step by step. Let's start by noting down all the given information clearly. Jan's current age is 30. That's a concrete number, so that's a good starting point.

Next, the problem mentions something about two years ago. So, I should probably calculate Jan's age two years ago. Since Jan is 30 now, two years ago she was $30 - 2 = 28$ years old. That seems straightforward.

Now, the problem says that two years ago, Mark was 5 years older than half of Jan's age at that time. So, half of Jan's age two years ago is $28/2 = 14$. Then adding 5 years to that gives $14 + 5 = 19$. So, Mark was 19 years old two years ago. Therefore, Mark's current age would be $19 + 2 = 21$ years old. Wait, that seems a bit young, but maybe that's correct based on the given info.

Then, the problem states that Jean is two years older than Mark. If Mark is currently 21, then Jean would be $21 + 2 = 23$ years old. But wait, that seems like a possible answer, but I need to check if I did all the steps correctly.
Let me verify each step again. Jan is 30 now. Two years ago, she was 28. Half of 28 is 14. Adding 5 gives 19, which is Mark's age two years ago. Adding 2 years brings Mark's current age to 21. Jean is two years older than Mark, so $21+2 = 23$. That seems consistent.

Feedback: The original plan is a good start but lacks specific details on how to calculate Mark's age two years ago and how to derive Jean's age from Mark's current age. To improve, the plan should explicitly outline the mathematical operations needed, such as calculating half of Jan's current age, adding 5 years, adjusting for the two-year time difference, and then applying the age difference between Jean and Mark. Additionally, the plan should emphasize verifying each step for consistency with the given information.

The Refined Plan is:

1. Identify known values.

2. Calculate Jan's age two years ago.

3. Determine Mark's age two years ago.

4. Calculate Mark's current age.

5. Find Jean's current age.

**Solver:**
According to the provided plan, to solve the problem step-by-step: Jan is currently 30, so two years ago Jan was $30 - 2 = 28$. Half of Jan's age two years ago is $28/2 = 14$, and Mark was 5 years older than that at the time, so Mark was $14 + 5 = 19$ two years ago and is therefore $19 + 2 = 21$ now. Since Jean is 2 years older than Mark, Jean is $21 + 2 = 23$ years old. Hence, the final answer is $\boxed{23}$.

## J. Case Study on LatentMAS

---

**Case Study (TextMAS on GSM8K)**

**Input Question:** Debra is monitoring a beehive to see how many bees come and go in a day. She sees 30 bees leave the hive in the first 6 hours, and then she sees 1/2 that many bees return in the next 6 hours. She sees two times as many bees as she saw first leave the hive fly from the hive and leave in the next 6 hours. Then every bee that left before that hadn't already returned returns to the hive in the next 6 hours. How many bees did Debra see return to the hive in the last 6 hours of the day?

---

**TextMAS Output**

**Planner Agent:** Okay, let's see. I need to figure out how many bees Debra saw return to the hive in the last 6 hours. Let me break down the problem step by step. First, the problem mentions different time intervals: first 6 hours, next 6 hours, then the next 6 hours, and finally the last 6 hours. So there are four 6-hour periods in a day, making up 24 hours total. Let me note each period and what happens in each. Starting with the first 6 hours: Debra sees 30 bees leave the hive. Let me write that down as the first data point. So, Period 1 (0-6 hours): 30 bees leave. Then, the next 6 hours (Period 2: 6-12 hours), she sees half that number return. Half of 30 is 15. So, 15 bees return in Period 2. Next, the third period (Period 3: 12-18 hours), she sees two times as many bees as she saw first leave the hive fly from the hive and leave. The first time she saw 30 bees leave, so two times that is 60. So, 60 bees leave in Period 3. Then, the fourth period (Period 4: 18-24 hours), every bee that left before that hadn't already returned returns to the hive.

**Critic Agent:** The reasoning plan correctly identifies all four 6-hour periods and reaches the right final conclusion, but the intermediate analysis contains several issues that reduce clarity and efficiency. First, the interpretation of the phrase "before that" is revisited multiple times, even though it simply refers to the event in the previous sentence (the 60 bees leaving), meaning only bees that left in Period 1 are relevant. Second, the plan temporarily entertains the possibility that bees may have left in Period 2, despite the problem statement explicitly describing only returns during this interval. Third, the plan repeatedly questions whether the 15 returning bees in Period 2 originate from the initial 30, even though the text directly states they are "half that many" and therefore unambiguously tied to Period 1. These detours introduce redundancy and confusion. Overall, while the conclusion of 15 returning bees is correct, the path to it is unnecessarily convoluted and would benefit from a more concise timeline-based analysis.

**Refiner Agent:** I will provide a refined plan based on previous agents' information.
1. Identify the Time Periods and Events. We divide the day into four 6-hour intervals and record the events in each:

- Period 1 (0–6 hours): 30 bees leave the hive.

- Period 2 (6–12 hours): Half of those bees return, i.e., 15 bees.

- Period 3 (12–18 hours): Twice the number of bees that left in Period 1 leave the hive: $2 \times 30 = 60$ bees.

- Period 4 (18–24 hours): Every bee that left *before Period 3* and has not yet returned comes back to the hive.

2. Track Bees Leaving and Returning.

- Period 1: 30 bees leave.

- Period 2: 15 of these 30 bees return, leaving 15 still outside the hive.

- Period 3: 60 new bees leave (these are not relevant for the final return count).

- Period 4: Only bees that left *before Period 3* and have not returned will come back.

3. Determine Which Bees Return in Period 4. The only bees that left before Period 3 are the 30 from Period 1. Of these, 15 returned in Period 2. Thus, the remaining 15 bees from Period 1 return in Period 4.
4. Return the Final Answer.

**Solver Agent:** Okay, let's see. I need to figure out how many bees Debra saw return to the hive in the last 6 hours. Let me break down the problem step by step.
First, the problem mentions different time intervals: first 6 hours, next 6 hours, then the next 6 hours, and finally the last 6 hours. So there are four 6-hour periods in a day, making up 24 hours total. Let me note each period and what happens in each. Starting with the first 6 hours: Debra sees 30 bees leave the hive. Let me write that down as the first data point. So, Period 1 (0-6 hours): 30 bees leave. Then, the next 6 hours (Period 2: 6-12 hours), she sees half that number return. Half of 30 is 15. So, 15 bees return in Period 2. Next, the third period (Period 3: 12-18 hours), she sees two times as many bees as she saw first leave the hive fly from the hive and leave. The first time she saw 30 bees leave, so two times that is 60. So, 60 bees leave in Period 3. Then, the fourth period (Period 4: 18-24 hours), every bee that left before that hadn't already returned returns to the hive. So, bees that left in Period 1 and Period 2. But some of those bees may have already returned in Period 2.
### Final Answer
Debra saw $\boxed{15}$ bees return to the hive in the last 6 hours of the day. (✗ *Wrong Final Answer*)

---

---

**Case Study (LatentMAS on GSM8K)**

**Input Question:** Debra is monitoring a beehive to see how many bees come and go in a day. She sees 30 bees leave the hive in the first 6 hours, and then she sees 1/2 that many bees return in the next 6 hours. She sees two times as many bees as she saw first leave the hive fly from the hive and leave in the next 6 hours. Then every bee that left before that hadn't already returned returns to the hive in the next 6 hours. How many bees did Debra see return to the hive in the last 6 hours of the day?

---

**LatentMAS Output**

We are given a sequence of events involving bees leaving and returning to the hive over the course of a day, broken into 6-hour intervals.

According to the refiner's plan, we need to track how many bees leave and return during each 6-hour interval, noting that 30 leave first, 15 return next, and 60 more leave afterward. In the final interval, all bees that had left and not yet returned came back, giving a total of 15 remaining from the first group, plus 60 from the third group, for a total of 75 returning bees.

Let's break it down step by step based on the plan:

**Step 1: Bees that leave in the first 6 hours.** Debra sees 30 bees leave the hive during the first 6-hour period.

**Step 2: Bees that return in the next 6 hours.** In the second 6-hour period, she sees half as many bees return: $\frac{1}{2} \times 30 = 15$ bees return.

**Step 3: Bees that leave in the third 6-hour period.** In the next 6 hours, twice the number from Step 1 leave: $2 \times 30 = 60$ bees leave.

**Step 4: Bees that return in the final 6 hours.** During the final 6 hours, every bee that had left earlier and had not yet returned comes back to the hive.
We track the bees that left and determine which have already returned:

- From the first 6 hours: 30 bees left; 15 returned in Step 2, leaving 15 bees still out.

- From the third 6 hours: 60 bees left; none have returned yet.

Thus, the total number of bees returning in the final 6 hours is $15 + 60 = 75$ bees.

**###Final Answer:** $\boxed{75}$ (✓ *Correct Final Answer*)

---

To comprehensively learn how latent collaboration changes multi-agent reasoning dynamics, we conduct a detailed case study on GSM8K using the Qwen3-14B backbone under the Sequential MAS setting. As shown in the example, TextMAS agents rely on lengthy textual exchanges that often amplify early reasoning errors, misinterpretations by the planner propagate through the critic and refiner, ultimately constraining the solver's search space. In contrast, LatentMAS operates entirely through latent working-memory transfer: each agent receives rich, continuous representations of prior reasoning rather than brittle text, enabling later agents to reinterpret, refine, and correct upstream reasoning without inheriting surface-level mistakes. This latent collaboration leads to more coherent intermediate steps, more stable numerical reasoning, and ultimately yields the correct final answer, where TextMAS fails. The case study illustrates how LatentMAS mitigates error compounding in multi-agent pipelines and demonstrates the qualitative advantage of latent over text-based communication.

## K. Prompt Template for LatentMAS

---

**Sequential LatentMAS Prompts on Numeric Tasks (GSM8K / AIME2024 / AIME2025)**

**System Prompt for All Agents:**
You are Qwen, created by Alibaba Cloud. You are a helpful assistant.

**Prompt for Planner Agent:**
You are a Planner Agent. Given an input question, design a clear, step-by-step plan for how to solve the question. Question: {question}
Your outlined plan should be concise with a few bulletpoints for each step. Do not produce the final answer. Now output your plan to solve the question below:

**Prompt for Critic Agent:**
You are a Critic Agent to evaluate the correctness of the input plan for the given question and provide helpful feedback for improving the plan. Question: {question}
The plan information is provided in latent KV representation format. Review the plan and question and output: (1) original plan contents (2) constructive feedback on the original plan. Format your response as follows: Original Plan: [Copy the provided Planner Agent's plan here] Feedback: [Your detailed feedback to improve the plan here] Now, output your response below:

**Prompt for Refiner Agent:**
You are a Refiner Agent to provide a refined step-by-step plan for solving the given question. Question: {question}
You are provided with: (1) latent-format information: a previous plan with feedback (2) text-format information: the input question you need to solve. Based on the input, write a refined and improved plan to solve the question. Make sure your output plan is correct and concise. Now, output your refined plan below:

**Prompt for Solver Agent:**
You are a helpful assistant. You are provided with latent information for reference and a target question to solve. Target Question: {question}
The latent information might contain irrelevant contents. Ignore it if it is not helpful for solving the target question. Now, reason step by step and output the final answer inside \boxed{YOUR_FINAL_ANSWER}:

---

**Sequential LatentMAS prompts for multiple-choice tasks (ARC-E, ARC-C, GPQA, MedQA)**

**System Prompt for All Agents:**
You are Qwen, created by Alibaba Cloud. You are a helpful assistant.

**Prompt for Planner Agent:**
You are a Planner Agent. Given an input question, design a clear, step-by-step plan for how to solve the question. Question: {question}
Your outlined plan should be concise with a few bulletpoints for each step. Do not produce the final answer. Now output your plan to solve the question below:

**Prompt for Critic Agent:**
You are a Critic Agent to evaluate the correctness of the input plan for the given question and provide helpful feedback for improving the plan. Question: {question}
The plan information is provided in latent KV representation format. Review the plan and question and output: (1) original plan contents (2) constructive feedback on the original plan. Format your response as follows: Original Plan: [Copy the provided Planner Agent's plan here] Feedback: [Your detailed feedback to improve the plan here] Now, output your response below:

**Prompt for Refiner Agent:**
You are a Refiner Agent to provide a refined step-by-step plan for solving the given question. Question: {question}
You are provided with: (1) latent-format information: a previous plan with feedback (2) text-format information: the input question you need to solve. Based on the input, write a refined and improved plan to solve the question. Make sure your output plan is correct and concise. Now, output your refined plan below:

**Prompt for Solver Agent:**
You are a helpful assistant. You are provided with latent information for reference and a target question to solve. Target Question: {question}
The latent information might contain irrelevant contents. Ignore it if it is not helpful for solving the target question. Your final answer must be selected from A,B,C,D. For example \boxed{A}. Do not add any other contents inside the box. Now, reason step by step and output the final answer inside \boxed{YOUR_FINAL_ANSWER}:

---

---

### Sequential LatentMAS prompts for python coding tasks (MBPP-Plus, HumanEval-Plus)

**System Prompt for All Agents:**
You are Qwen, created by Alibaba Cloud. You are a helpful assistant.

**Prompt for Planner Agent:**
You are a Planner Agent. Given an input question, design a clear, step-by-step plan for how to solve the question. Question: {question}
Your outlined plan should be concise with a few bulletpoints for each step. Do not produce the final answer. Now output your plan to solve the question below:

**Prompt for Critic Agent:**
You are a Critic Agent to evaluate the correctness of the input plan for the given question and provide helpful feedback for improving the plan. Question: {question}
The plan information is provided in latent KV representation format. Review the plan and question and output: (1) original plan contents (2) constructive feedback on the original plan. Format your response as follows: Original Plan: [Copy the provided Planner Agent's plan here] Feedback: [Your detailed feedback to improve the plan here] Now, output your response below:

**Prompt for Refiner Agent:**
You are a Refiner Agent to provide a refined step-by-step plan for solving the given question. Question: {question}
You are provided with: (1) latent-format information: a previous plan with feedback (2) text-format information: the input question you need to solve. Based on the input, write a refined and improved plan to solve the question. Make sure your output plan is correct and concise. Now, output your refined plan below:

**Prompt for Solver Agent:**
You are a helpful assistant. You are provided with latent information for reference and a target question to solve. Target Question: {question}
The latent information might contain irrelevant contents. Ignore it if it is not helpful for solving the target question. Your must reason step-by-step to solve the **provided Target Question** witout outputing other irrelevant inforamtion. You must put all python code as self-contained Python function in markdown code blocks. For example
```python
import math
def add(a, b):
return a + b```.
Do not add any other contents inside the markdown code block. Now, reason step by step and output the final answer inside ```python YOUR_PYTHON_CODE```:

---

### Hierarchical LatentMAS prompts for numeric-answer tasks (GSM8K, AIME2024, AIME2025)

**System prompt for All Agents:**
You are Qwen, created by Alibaba Cloud. You are a helpful assistant.

**Prompt for Math Agent:**
You are a math agent. Given the input question, reason step-by-step and put the final answer inside \boxed{YOUR_FINAL_ANSWER}.
Question: {question}
Your response:

**Prompt for Science Agent:**
You are a science agent. Given the input question, reason step-by-step and put the final answer inside \boxed{YOUR_FINAL_ANSWER}. Question: {question}
Your response:

**Prompt for Code Agent:**
You are a code agent. Given the input question, reason step-by-step and put the final answer inside \boxed{YOUR_FINAL_ANSWER}.
Question: {question}
Your response:

**Prompt for Task Summarizer Agent:**
You are a task summarizer. Given the input question and responses from previous agents as reference, reason step-by-step and put the final answer inside \boxed{YOUR_FINAL_ANSWER}.
Question: {question}
Your response:

**Hierarchical LatentMAS prompts for multiple-choice tasks (ARC-E, ARC-C, GPQA, MedQA)**

**System Prompt for All Agents:**
You are Qwen, created by Alibaba Cloud. You are a helpful assistant.

**Prompt for Math Agent:**
You are a math agent. Given the input question, reason step-by-step and put the final answer inside \boxed{YOUR_FINAL_ANSWER}. Your final answer must be selected from A,B,C,D. For example \boxed{A}. Do not add any other contents inside the box. Question: {question}
Your response:

**Prompt for Science Agent:**
You are a science agent. Given the input question, reason step-by-step and put the final answer inside \boxed{YOUR_FINAL_ANSWER}. Your final answer must be selected from A,B,C,D. For example \boxed{A}. Do not add any other contents inside the box. Question: {question}
Your response:

**Prompt for Code Agent:**
You are a code agent. Given the input question, reason step-by-step and put the final answer inside \boxed{YOUR_FINAL_ANSWER}. Your final answer must be selected from A,B,C,D. For example \boxed{A}. Do not add any other contents inside the box. Question: {question}
Your response:

**Prompt for Task Summarizer Agent:**
You are a task summarizer. Given the input question and responses from previous agents as reference, reason step-by-step and put the final answer inside \boxed{YOUR_FINAL_ANSWER}. Your final answer must be selected from A,B,C,D. For example \boxed{A}. Do not add any other contents inside the box. Question: {question}
Your response:

---

**Hierarchical LatentMAS prompts for python coding tasks (MBPP-Plus, HumanEval-Plus)**

**System Prompt for All Agents:**
You are Qwen, created by Alibaba Cloud. You are a helpful assistant.

**Prompt for Math Agent:**
You are a math agent. Given the input question, reason step by step and provide an efficient and self-contained Python function that solves the following problem in a markdown code block. You must put all python code as self-contained Python function in markdown code blocks. For example
```python
import math
def add(a, b):
return a + b```. Do not add any other contents inside the markdown code block.
Question: {question}
Your response:

**Prompt for Science Agent:**
You are a science agent. Given the input question, reason step by step and provide an efficient and self-contained Python function that solves the following problem in a markdown code block.
You must put all python code as self-contained Python function in markdown code blocks. For example
```python
import math
def add(a, b):
return a + b```. Do not add any other contents inside the markdown code block.
Question: {question}
Your response:

**Prompt for Code Agent:**
You are a code agent. Given the input question, reason step by step and provide an efficient and self-contained Python function that solves the following problem in a markdown code block.
You must put all python code as self-contained Python function in markdown code blocks. For example
```python
import math
def add(a, b):
return a + b```. Do not add any other contents inside the markdown code block.
Question: {question}
Your response:

**Prompt for Task Summarizer Agent:**
You are a task summarizer. Given the input question and responses from previous agents as reference, reason step by step and provide an efficient and self-contained Python function that solves the following problem in a markdown code block.
You must put all python code as self-contained Python function in markdown code blocks. For example
```python
import needed_library
def FUNC_NAME(a, b):
return a + b```. Do not add any other contents inside the markdown code block.
Input Question: {question}
Your response:

