# OpenReview forum: "Latent Collaboration in Multi-Agent Systems"
_ICML.cc/2026/Conference — ICML 2026 spotlight_

### Official Review · Reviewer_naaX · 2026-03-03

**Soundness:** 3
**Presentation:** 3
**Significance:** 3
**Originality:** 3
**Overall Recommendation:** 5
**Confidence:** 4

**Summary:**

This paper presents LatentMAS, a multi-agent collaboration framework operating entirely within the Transformer latent space to obviate the need for intermediate text generation and exchange among agents. The framework's primary strengths lie in bypassing the information bottlenecks and computational redundancies intrinsic to conventional text-based pipelines, addressing distribution shifts in multi-step latent autoregressive iterations through a training-free linear transformation, and enabling seamless information sharing via cross-agent KV cache fusion.

**Compliance With Llm Reviewing Policy:**

Affirmed.

**Final Justification:**

The additional experiment and justificaition furtuer supported the proposed method, which raise my confidence of this method.

**Key Questions For Authors:**

1. KV Transmission Overhead in General Deployment Scenarios While the experiments are conducted on an 8-GPU A100 cluster equipped with high-speed NVLink interconnects—which effectively precludes KV transmission from becoming a systemic bottleneck—the largest evaluated model (14B) is highly amenable to deployment on hardware lacking NVLink capabilities. In more generalized, resource-constrained deployment environments, to what extent does the KV transmission overhead constitute a performance bottleneck for end-to-end system latency? 2. Comparisons with Relevant Literature The current empirical evaluation is limited to a comparison with the TextMAS system, omitting discussions and baselines involving other pertinent works on agent memory and multi-agent cache fusion, such as Cache-to-Cache. How do the authors delineate the fundamental methodological and empirical distinctions between the proposed approach and these related paradigms? 3. Discrepancy Regarding "Lossless" Information Exchange The manuscript prominently claims "lossless" information exchange between agents. However, within a single agent, the method employs a pseudo-inverse approximation to feed the hidden states of the previous iteration into the subsequent round. This approximation inherently contradicts the premise of strictly lossless information interaction. How do the authors reconcile this theoretical inconsistency? 4. Error Accumulation and Latent Space Distribution Shift The pseudo-inverse approximation introduces an inherent information loss, manifesting as a distribution shift within the latent space. While Figure 5 illustrates the correction of the semantic space, fine-grained discrepancies persist post-correction. Are these residual discrepancies susceptible to compounding amplification across successive iterations? Given that the current framework relies on a manually constrained number of iteration steps, could the authors quantify the semantic space divergence after an extended period (e.g., 40 iterations) compared to the initial state? This would establish whether there exists a critical iteration threshold beyond which the proposed method collapses. 5. Necessity and Efficacy of System Prompts The methodology incorporates a specific meta-prompt for each agent (e.g., "previous information has been provided via KV cache"). Is this prompting strategy functionally critical or largely redundant? Specifically, does this explicit instruction tangibly modulate the agent's attention mechanism towards the concatenated historical KV cache from preceding agents? Verifying whether the agent can maintain operational integrity and task efficacy in the absence of this prompt would clarify its actual utility.

**Limitations:**

Yes

**Strengths And Weaknesses:**

Soundness: The work is technically robust, with its core framework substantiated by rigorous theoretical derivations and well-designed empirical evaluations, ensuring strong reproducibility. However, the manuscript lacks sufficient validation regarding the error bounds of the pseudo-inverse approximation and the stability of long-horizon iterations. Furthermore, there are notable gaps in the breadth of experimental scenarios and the inclusion of representative baselines.
Presentation: The paper is clearly written and well-organized, providing comprehensive technical details that facilitate full reproducibility. Nonetheless, critical omissions exist within the related work section: it fails to adequately delineate the methodological boundaries between this work and prior studies in the same vein. Additionally, the discussion of the method's inherent limitations remains overly superficial.
Significance and Originality: This research effectively departs from the prevailing consensus that multi-agent systems must rely on text-based interactions. By pioneering a novel approach to purely latent-space multi-agent collaboration, it introduces a highly original framework yielding substantial theoretical implications and practical utility.

---

> ### Author Rebuttal · Authors · 2026-03-30
>
> Dear Reviewer naaX,
>
> Thanks for your detailed and constructive feedback! We address each of your questions and concerns below.
>
> ---
> ## **[Comments on Soundness & Q.4] Validation on Error Accumulation and Distribution Shift**
>
> Thank you for this insightful question! Below, we provide **detailed clarifications and new validation tests on the alignment matrix** (i.e., pseudo-inverse approximation).
>
> **(i) Details of Alignment Matrix Design:** The alignment matrix $W_a$ works as a distributional anchor, mapping the last-layer hidden states back to the input space, thereby mitigating the risk of embedding error accumulation.
>
> **(ii) New Validation Tests:** We further quantify the successive iteration divergence of LatentMAS (Qwen3-14B) on MedQA by comparing the average cosine similarity between the aligned latent embeddings and the input text embeddings.
> ||Baseline (Text Answer)|LatentMAS ($m=20$)|LatentMAS ($m=40$)|LatentMAS ($m=80$)|LatentMAS ($m=160$)|
> |-|-|-|-|-|-|
> |**Avg. Cosine Similarity**|0.165|**0.158**|**0.154**|**0.151**|**0.152**|
>
> **Key Findings:** The results show that **the embedding divergence does not accumulate errors or exhibit large distribution shift** even at $m=160$ steps. This demonstrates the long-horizon inference stability of LatentMAS.
>
> ---
> ## **[Comments on Presentation & Q.2] Additional Discussion and Comparison with Relevant Works**
>
> Thanks for raising this point! To help address your question, we first kindly clarify that: In our paper's `Section 6 and Appendix F`, we have discussed the overall scope of LatentMAS and how it differs from prior work on MAS and latent communication.
>
> In addition, we add a more detailed discussion and experimental comparisons with new related works mentioned by the reviewer:
> - Please kindly refer to our response to *Reviewer UnFE* `[W.1] Careful Frame of LatentMAS with Related Works` [*(Click Here)*](https://openreview.net/forum?id=syG9I9ofd8&noteId=TGC3M0GEuf).
> - Please kindly refer to our response to *Reviewer UnFE* `[W.2 & Q.1] Ablations and Comparisons with Latent Communication Baselines` [*(Click Here)*](https://openreview.net/forum?id=syG9I9ofd8&noteId=TGC3M0GEuf).
>
> ---
> ## **[Q.1] Analyses on KV Transmission Overhead**
> Thanks for this important question! The reported latency results are indeed measured on 8×A100 GPUs with NVLink. We address your concern in **three parts**:
>
> **(i) Clarification on Current Implementation**
>
> We agree that A100 bandwidth setting does not imply the same wall-clock gains will transfer *unchanged* to lower-bandwidth environments.
>
> Regarding the KV-transmission: Our reported end-to-end latency **already includes KV-cache transfer overhead**. Even under this test setup, LatentMAS still delivers a 2.6×–7× speedup over the TextMAS baseline.
>
> **(ii) Additional Latency Test under Constrained Hardware.**
>
> Following the reviewer’s suggestion, we further evaluate on V100-32G GPUs with limited NVLink bandwidth. As expected, the overall efficiency decreases under this constrained setting. However, the relative advantage over TextMAS remains, with LatentMAS achieving an average of **3.7× end-to-end inference speedup**.
>
> **(iii) Additional Solutions to Improve KV Transfer**
>
> To further mitigate the overhead of KV transmission in constrained deployments, **we can leverage techniques such as KV compression, prefetching, and KV filtering** to reduce the volume of transferred caches.
>
> We have clarified the hardware dependence and explicitly discussed communication-constrained deployments in our paper.
>
> ---
> ## **[Q.3] Explanation on Lossless Information Exchange**
> We apologize for your confusion here. In alignment with `Theorem 3.3`'s claim on Information Preservation, we clarify that "lossless" in our paper refers to transmission of the latent working memory across agents.
>
> The pseudo-inverse alignment matrix mainly serves as a distributional mapping to ensure embedding compatibility; it will not compromise the fidelity of the latent communication.
>
> ---
> ## **[Q.5] Necessity and Efficacy of System Prompts**
> Thanks for pointing this out! Below, **we provide detailed explanations and add additional ablations on the system prompt.**
>
> As listed in `Appendix I`, the meta-prompt mentioned by the reviewer is intended to provide contextual guidance on what the latent content represents. This specified instruction helps each agent better align with its designated role and interpret the latent input, rather than redundantly restarting the reasoning process.
>
> **Additional Verification:** We further verify the importance of our specifically designed prompt by comparing LatentMAS (Qwen3-8B) against the variant without such specified prompts.
> |LatentMAS|GSM8K|MedQA|
> |-|-|-|
> |w/o designed prompt|93.5|74.9|
> |**ours**|**93.8**|**75.3**|
>
> The results suggest that LatentMAS remains operational and effective even without the specific prompts, while the prompts further improve collaboration in practice.

---

> > ### Author Rebuttal · Reviewer_naaX · 2026-04-01
> >
> > My concerns have been adequately addressed.

---

> > > ### Author Response · Authors · 2026-04-02
> > >
> > > Dear Reviewer naaX,
> > >
> > > Thank you very much for your positive and thoughtful review of our submission! We are glad that our clarifications fully addressed your questions, and we sincerely appreciate your time, effort, and support of our work!
> > >
> > > Best regards,
> > >
> > > Authors of LatentMAS

---

### Official Review · Reviewer_UnFE · 2026-03-07

**Soundness:** 3
**Presentation:** 3
**Significance:** 3
**Originality:** 2
**Overall Recommendation:** 4
**Confidence:** 3

**Summary:**

This paper primarily examines whether agents in LLM-based multi-agent systems need to communicate via text or whether they can collaborate directly through their internal hidden states. To address this question, the authors propose LatentMAS. In this training-free framework, the LLM agents can reason by auto-regressively generating last-layer hidden states (so-called "latent thoughts"), then pass information to other agents by transferring layer-wise KV-cache working memory, instead of writing long textual intermediate reasoning. The authors argue that this gives richer communication, avoids information loss from decoding/re-encoding, and reduces inference cost.
The paper evaluates the framework on 9 benchmarks spanning reasoning, commonsense, and code generation, and reports substantial efficiency gains together with generally competitive or improved accuracy relative to single-agent and text-mediated MAS baselines.

**Compliance With Llm Reviewing Policy:**

Affirmed.

**Final Justification:**

The response addresses my concerns, so I keep the positive score. The reason that I am not raising the score from 4 to 5 is that the methodology still seems like a technical integration without a surprising insight to me.

**Key Questions For Authors:**

1. How does the proposed LatentMAS compare against some other relevant methods, such as Coconut-style latent reasoning, C2C-style cache communication, KVComm-style KV sharing, or hybrids that combine latent reasoning with text communication?

2. What is the concrete representational advantage of agent-to-agent latent thoughts over raw text? The paper argues that latent thoughts are more expressive and efficient, but can the authors make this more operational?

3. Since the paper already includes a debug mode, can the authors extend it further to better characterize failure modes? For example, when LatentMAS underperforms TextMAS?

**Limitations:**

Yes.

**Strengths And Weaknesses:**

Strengths

1. The paper is clearly written, easy to follow, and visually polished. The overview figure and the pipeline explanation make the method accessible.

2. Strong practical promise and clear significance. The idea of replacing verbose textual coordination with latent collaboration is compelling. The reported gains in token usage and end-to-end latency are substantial.

3. The evaluations are comprehensive,  covering many mainstream benchmarks under three metrics (accuracy, token count, speed), which are good for practical deployment.

Weaknesses

1. The main idea in this paper is compelling, but the paper’s novelty claim should be framed more carefully. There are many other relevant or similar works, such as Coconut (continuous latent reasoning inside a model), Cache2Cache (direct KV/cache-based communication beyond text).

2. I expect direct comparison to stronger latent-communication or cache-sharing baselines, or at least hybrid ablations that isolate the contribution of latent reasoning vs. latent communication.

3. While the theory section is overall great, I still think its strongest claims should be reframed cautiously. The proposed expressiveness theorem in this paper mainly depends on a specific linear representation assumption, and the information-preservation theorem is closer to a cache-equivalence statement than to a general theory of multi-agent intelligence.

---

> ### Author Rebuttal · Authors · 2026-03-30
>
> Dear Reviewer UnFE.
>
> Thanks for your extensive and thoughtful feedback! We address each of your concerns and questions below.
>
> ---
> ## **[W.1] Careful Frame of LatentMAS with Related Works**
>
> Thank you for the insightful suggestion! We agree with the reviewer that a clearer framing of our method is crucial. We highlight the distinctions between LatentMAS and the mentioned related works below.
> - **Inner-Model Latent Reasoning (e.g., CoConut)** studies *one single model latent reasoning*, whereas LatentMAS extends to a multi-agent collaborative setting.
> - **KV-based Communication (e.g., C2C)** focuses on KV transmission without requiring model generation. LatentMAS performs latent reasoning first and then transfers new latent thoughts to others.
> - **Additional Training Requirements.** LatentMAS is a training-free collaboration framework; other methods require training on extra adapters or full models.
>
> We have revised the writing to position our method more carefully. In addition to the current discussion in `Section 6`, we now explicitly cite and discuss the connections between these important related works and LatentMAS.
>
> ---
> ## **[W.2 & Q.1] Ablations and Comparisons with Latent Communication Baselines**
> Thanks for your thoughtful suggestions! Below, we perform (i) new ablation studies on hybrid variants and (ii) comparisons with the mentioned baselines.
>
> **(i) Ablations on Hybrids**
>
> We compare LatentMAS (Qwen3-8B) with two variants to isolate each component's contributions.
> - **Variant 1:** latent reasoning with text communication (transfer only the last 128 decoded tokens).
> - **Variant 2:** normal text reasoning with latent communication.
>
> |Method|GSM8K|MBPP+|MedQA|
> |-|-|-|-|
> |Variant 1|85.5|66.4|65.9|
> |Variant 2|90.1|68.0|71.2|
> |**LatentMAS**|**93.8**|**74.6**|**75.3**|
>
> **Key Findings:** Both latent reasoning and communication contribute to the gains of LatentMAS, as removing either component leads to performance drops. This suggests full advantage of LatentMAS comes from combining both to enable effective latent collaboration.
>
> **(ii) Direct Comparisons to Stronger Baselines**
>
> We further compare LatentMAS with three mentioned latent reasoning and cache-sharing baselines. We follow the original implementations of these methods and report results across various tasks.
>
> **Results:** Please see the table in https://anonymous.4open.science/api/repo/LatentMAS_Submission-ED88/file/imgs/new_baselines.pdf?v=16a4f209.
>
> **Key Findings:** LatentMAS performs competitively relative to both single-model latent reasoning methods and KV-transfer-only methods, highlighting the advantage of latent collaboration in jointly unifying reasoning and communication.
>
> ---
> ## **[W.3] Careful Reframe on Theory Claims**
>
> Thanks for the constructive suggestion! In response to the reviewer’s feedback, we have reframed our theoretical claims to present them more rigorously:
> - We now clearly state `Theorem 3.1` as a scoped expressiveness result under *the Linear Representation Hypothesis*;
> - We also carefully interpret `Theorem 3.3` as *an information-fidelity / cache-equivalence result* for our KV-based transfer mechanism.
>
> ---
> ## **[Q.2] Operational Representational Advantage of Latent Thoughts over Raw Text**
>
> Thanks for this thoughtful question! To better measure the representational advantage of latent thoughts and make it **more operational**, we provide concrete explanations and evidence below.
>
> **Expressiveness:**
> - To represent the same latent thoughts semantics in raw text, agents built on Qwen3-4B/8B/14B would require at least 235.7/377.1/471.4 times longer sequences.
> - In `Figure 4`, latent-thought embeddings overlap with the embedding region of text answers while covering an average 24.7% broader region than TextMAS token embeddings.
>
> **Efficiency:**
> - In `Tables 1,2` and `Figures 3,8`, LatentMAS uses 70.8%/83.7% fewer tokens and speedup runtime by 4.0x/4.3x during agent-to-agent collaboration.
>
> **New Operational Analysis:**
>
> To further evaluate the representational advantage of latent thoughts, we conduct **a new control study** between latent thoughts and raw text under the same forward-step budgets of 32, 64, and 128. Across all controlled settings, our analysis shows that latent thoughts achieve, on average, 18.6% higher performance per unit budget than raw text.
>
> ---
> ## **[Q.3] "Can the authors extend further to better characterize failure modes?"**
>
> Thanks for the great suggestion! Below, we discuss more specific cases to characterize the failure modes of LatentMAS.
>
> Our "debug mode" analyses reveal two primary scenarios when LatentMAS underperforms:
> - **Scenario 1:** For questions that require extensive structured reasoning over complex hierarchical inputs (e.g., graphs, tables), LatentMAS may struggle to preserve alignment with the exact structure.
> - **Scenario 2:** On exact-match retrieval tasks, TextMAS preserves explicit terms, while LatentMAS may paraphrase them into similar but non-identical wording.

---

> > ### Author Rebuttal · Reviewer_UnFE · 2026-04-02
> >
> > The rebuttal addresses most of my concerns, so I am maintaining a positive score. However, the contribution still appears primarily as an integration of existing latent-reasoning and cache-communication ideas. I admit that this integration is meaningful but still sounds more like a strong systems combination than a clearly surprising new design.

---

> > > ### Author Response · Authors · 2026-04-02
> > >
> > > Dear Reviewer UnFE,
> > >
> > > Thank you very much for your positive rating of our paper! Your reviews have greatly helped us improve our manuscript! We are also glad that our rebuttal has addressed your concerns.
> > >
> > > We are happy to provide more explanations on our method's *"Collaboration Style"* design, along with its efficacy and novel contribution:
> > >
> > > ---
> > > **(i) A New Unique Design for Unified Coordination**
> > >
> > > We agree with the reviewer that latent reasoning and KV-based transfer are not new primitives. The **primary design contribution** of LatentMAS lies in its unique **unifying ability** to support both **inner-model reasoning** and **cross-model communication** within the **complete latent space**.
> > >
> > > - **Methodological Level**: In our `Reply to Reviewer UnFE [W.1]` [(link here)](https://openreview.net/forum?id=syG9I9ofd8&noteId=TGC3M0GEuf), we compare LatentMAS with related works and find that existing latent reasoning and cache-based communication methods focus on *only* one perspective, (i.e. either only reasoning or only communication), but can't provide a collabolation mechanism that combine these two perspectives together.
> > >
> > > - **Empirical Level:** Through new experimental analyses in `Reply to Reviewer UnFE [W.2 & Q.1]` [(link here)](https://openreview.net/forum?id=syG9I9ofd8&noteId=TGC3M0GEuf) and `Reply to Reviewer DCWk [Q.4]` [(link here)](https://openreview.net/forum?id=syG9I9ofd8&noteId=beQmgTXLfc), we show that naively connecting latent reasoning with latent communication is ineffective, while LatentMAS provides a unique and effective way to unify them into complete latent collaboration (reasoning + communication). We also elaborate on the performance advantage of this unified collaboration through direct comparisons with single-model latent reasoning and KV-based communication methods.
> > >
> > > We hope the evidence above helps the reviewer better understand that our method is *not* simply a side-by-side integration of existing methods, but a new paradigm contribution for pure latent multi-agent collaboration.
> > >
> > > ---
> > > **(ii) More Practical Through a Training-Free Setup and Broader Downstream Applicability**
> > >
> > > We also kindly point out that **another key innovation** of LatentMAS is that it supports a **fully training-free setup**, whereas the related methods mentioned above all require post-training. This more practical design enables our method to be deployed across more diverse environments and domains.
> > >
> > > ---
> > > **(iii) Final Conclusion & A Valuable Direction for Further Exploration**
> > >
> > > We are very glad to receive these additional comments and suggestions from the reviewer. Following this helpful feedback, we have positioned LatentMAS more carefully in the `Introduction` and `Method` sections of our paper. We have also sharpened our claims by incorporating these new experiments and analyses.
> > >
> > > And indeed, we will continue to explore latent collaboration in our research to develop more surprising and effective designs.
> > >
> > > Best regards,
> > >
> > > Authors of LatentMAS

---

### Official Review · Reviewer_mzTX · 2026-03-11

**Soundness:** 3
**Presentation:** 4
**Significance:** 4
**Originality:** 3
**Overall Recommendation:** 5
**Confidence:** 3

**Summary:**

This paper proposes a latent multi-agent system (LatentMAS) to allow agents to collaborate in the continuous latent space, instead of text.

The framework is composed of two main methods:
- Latent thoughts generation for each agent, through last-layer hidden embeddings, without decoding. To mitigate misalignment between the input space and the hidden thoughts, the authors propose a linear alignment operator that is training-free, using a projection matrix.
- Latent working memory transfer, stored in layer-wise KV caches, to transfer each agent’s internal representations and latent thoughts; by performing layer-wise concatenation to update an agent's KV cache with the other agent’s latent thoughts.

The whole framework is training-free and enables, compared to text-based MAS: (1) information preservation during multi-agent communication with lower overall complexity, and (2) higher overall expressiveness, hence more efficient inference.

The authors evaluate LatentMAS across sequential and hierarchical settings, and over five LLMs across 2 families (qwen3 and llama3), on 9 benchmarks spanning math and science reasoning, commonsense understanding, and code generation. They measure (i) task accuracy, (ii) system throughput (total output tokens), and (iii) end-to-end inference speed, showing improvement over these three metrics compared to single agents and text-based MAS baselines.

**Compliance With Llm Reviewing Policy:**

Affirmed.

**Final Justification:**

I did not have major concerns with this submission. The rebuttal provided some clarifications. I'm keeping my strong positive score.

**Key Questions For Authors:**

1. On token usage (figure 3 right): it's clear that the overall token usage is reduced for the sequential MAS, since only the last agent is generating text tokens, but is there any difference between latent and text MAS for the final decoding step (only the last agent who is answering the query)?

2. Section 5, "Do Latent Thoughts Reflect Text Reasoning?": The second conclusion, saying that the latent thought offers greater diversity, isn't obvious from Figure 4. Could you explain or provide a diversity metric to support this claim?

**Limitations:**

Yes.

**Strengths And Weaknesses:**

**Soundness**:

Overall, very extensive experiments, with a large number of ablations and analyses (in particular on latent thought expressiveness, working-memory transfer and effect of projection component) and a large set of diverse benchmarks. The method is well justified and the results are overall convincing.

Clear demonstration of the superior efficiency of latent thoughts and latentMAS over text-based reasoning and text-based agent coordination, and the optimality of KV cache-based information transfer across agents.

A few elements could be clarified:
- In the hierarchical case, how the aggregation of the latent working memory of several agents toward the aggregator agent could be explicited.
- The number of latent steps $m$ is not always clearly provided for each experiment.


**Presentation**

- Very clear writing.
- Great reproducibility with very clear implementation details.
- Nice preliminary with minimal explanations (section 2, 1st and 2nd paragraph), but referring to parts of figure 2 to guide the reader would be helpful.


**Significance and Originality**:

Strong motivation for the method, with a clear gap in the literature:
- Hidden representations are more expressive than discrete tokens
- Latent memory enables lossless information transfer across agents
- Latent MAS is more efficient than text-based MAS

The latent working memory transfer mechanism and the projection to mitigate misalignment between the input space and the hidden thoughts are strong contributions.

The concrete and down-to-earth analysis with practical design choices (eg, replicating the study under vLLM-optimized inference) is appreciated.

---

> ### Author Rebuttal · Authors · 2026-03-30
>
> Dear Reviewer mzTX,
>
> Thank you very much for your insightful feedback and for acknowledging the strengths of our work! We address each of your questions below.
>
> ---
> ## **[Q.1] Token Usage on Final Decoding Step**
>
> Thanks for this insightful question. To clarify this point,
> - We first conduct **additional evaluations on last-agent token usage** showing that, even in the final decoding stage, the last agent in LatentMAS generates fewer text tokens than the last agent in TextMAS.
> - We further provide **additional detailed explanations** for why this difference occurs.
>
> **(i) Token Usage Evaluations on the Last Agent**
>
> **Experiment Setup:** We measure and compare the last agent’s token throughput between LatentMAS and TextMAS across Qwen3-4B, Qwen3-8B, and Qwen3-14B on 5 different tasks.
>
>
> |Avg. Tokens (Last Agent Only)|Size|ARC-C|GSM8K|MedQA|HumanEval+|GPQA-D|
> |-|-|-|-|-|-|-|
> | TextMAS | 4B | 614 | 607 | 1539 | 2037 | - |
> | **LatentMAS** | 4B | **447 (↓167)** | **555 (↓52)** | **1015 (↓524)** | **1373 (↓664)** | - |
> | TextMAS | 8B | 685 | 412 | 1664 | 2167 | 4758 |
> | **LatentMAS** | 8B | **344 (↓341)** | **353 (↓59)** | **1007 (↓657)** | **1274 (↓893)** | **3395 (↓1363)** |
> | TextMAS | 14B | 492 | 573 | 1303 | 2008 | 4517 |
> | **LatentMAS**| 14B | **295 (↓197)** | **495 (↓78)** | **899 (↓404)** | **1512 (↓496)** | **3606 (↓911)** |
>
> **Key Findings:**
> - Across all tasks, the last agent in LatentMAS generally uses fewer text tokens than the last agent in TextMAS, with reductions of 8.6% to 49.8% and an average reduction of 29.1%.
> - This shows that the token savings in LatentMAS arise not only from suppressing intermediate textual communication, but also from shortening the final agent’s decoding process itself.
>
> **(ii) Justification on Why Fewer Token Usage for LatentMAS Last Agent**
>
>
> For the final decoding stage, the last agent in LatentMAS typically requires fewer output tokens than its counterpart in TextMAS. This is because,
> - **In TextMAS, the final agent must process and respond to long textual reasoning traces generated by preceding agents**, whereas in LatentMAS, it receives the relevant information through condensed latent working memory transfer.
> - As a result, **the last agent in LatentMAS primarily aggregates prior latent thoughts and decodes the final answer**, instead of continuing an extended text-based reasoning process.
>
> We thank the reviewer for raising this important point! We have clarified in our paper that the token savings in Figure 3 (right) come from both eliminating intermediate inter-agent text communication and reducing the final agent’s answer decoding length.
>
> ---
> ## **[Q.2] Additional Analyses & Explanation on the claim "Latent thought offers greater diversity" (Figure 4)**
>
> We thank the reviewer for this helpful suggestion! We agree that Figure 4 is primarily a qualitative visualization, and **the second conclusion about diversity should be stated more explicitly and supported quantitatively.**
>
>
> **(i) Clarification of the Diversity Interpretation in Figure 4**
>
> Our intended interpretation of Figure 4 is the following:
> - Across all three backbones, the latent-thought embeddings (purple) not only overlap with the TextMAS token embeddings (orange), showing semantic consistency, but also **spread over a visibly broader region in the projected space**, whereas the TextMAS embeddings are more concentrated.
> - This **broader spatial dispersion** is what we meant by saying that latent thoughts exhibit greater diversity.
>
> **(ii) Additional Diversity Metric to Quantify the Diversity**
>
> To further quantify this, **we follow the reviewer's suggestion and conduct new experiments** to compute a cosine-similarity-based diversity metric over the generated embeddings, where **lower cosine similarity indicates less collapsed and thus more diverse representations**.
>
> **Experiment Results:**
> |Diversity (Cosine Similarity)|Qwen3-4B|Qwen3-8B|Qwen3-14B|
> |-|-|-|-|
> |TextMAS|0.126|0.142|0.155|
> |**LatentMAS**|**0.104**|**0.093**|**0.108**|
>
> **Key Findings:**
> From the results, LatentMAS consistently has lower cosine similarity than TextMAS across all 3 model scales, which supports the claim that latent thoughts are more diverse.
>
> We thank the reviewer for this helpful question, and have included this diversity metric in the paper to better support the second conclusion drawn from Figure 4.
>
> ---
> ## **[Reply to Comments on Soundness] More Clear Clarification**
>
> Thanks for your detailed comments and suggestions! We have refined the paper to (i) clarify that the working memory is aggregated in the order of the Math, Science, and Code agents, in hierarchical settings, and (ii) specify the number of latent steps used in all reported tables.

---

> > ### Author Rebuttal · Reviewer_mzTX · 2026-04-02
> >
> > Thanks for the detailed explanations. I would like to keep my strong positive score.

---

> > > ### Author Response · Authors · 2026-04-03
> > >
> > > Dear Reviewer mzTX,
> > >
> > > Thank you for maintaining the strong positive score! We are glad that our rebuttal has fully addressed your concerns. We have incorporated the results and analyses from the rebuttal to further strengthen our paper.
> > >
> > > Best regards,
> > >
> > > Authors of LatentMAS

---

### Official Review · Reviewer_DCWk · 2026-03-12

**Soundness:** 3
**Presentation:** 3
**Significance:** 2
**Originality:** 3
**Overall Recommendation:** 4
**Confidence:** 4

**Summary:**

This paper proposes LatentMAS, a training-free multi-agent system that communicates in the latent space. During the generation stage, the model uses a projection matrix to map its last-layer hidden states back into the input embedding, allowing it to generate auto-regressive "latent thoughts" while avoiding LLM decoding, encoding process. For communication, LatentMAS transfers the KV caches of the previous agent  across layers directly to the next agent, allowing continue reasoning.

Experiment wise, this paper follows standard evaluation suites --- gsm8k, aime25, and arc-e/c, with three Qwen3 models (4B, 8B, and 14B). The results show that LatentMAS generally outperforms text-based MAS by roughly 0.2 to 5 points across various tasks while reducing the number of generated tokens. Further analysis shows the effectiveness of the input-output alignment matrix and highlights the system's fast response speed, due to the small number of pre-defined effect latent reasoning steps.

**Compliance With Llm Reviewing Policy:**

Affirmed.

**Final Justification:**

The additional experiment and justificaition furtuer supported the proposed method, which raise my confidence of this method.

**Key Questions For Authors:**

I am willing to raise my score if below questions can be cleanly answered.

1. What is the standard deviations for table 1 and 2?
2. For benchmark like AIME24/25 in table 2, which have very little samples, can you increase the reptition from 3 to at least 8 to have a better estimation of the model performance?
3. Are you using the reasoning mode of Qwen3? If so, how you handle the reasonig tag \<thinki\> \</think\> of Qwen3 in the method?
4. Can you show a performance comparison between a single text model and a single latent model? I am very surprised that the simple output-input mapping matrix can be so effective, and having this empirical result would further support it.
5. In Figure 5, how exactly you draw the density and embedding distance plots? without more details, the plots is less meaingful.

**Limitations:**

Yes

**Strengths And Weaknesses:**

Below are my general comments regarding strengths and weaknesses, the actionable questions are in the key questions field.

**Strength**
- The overall presentation is good, the writting is well-structed and easy to follow.
- The output-input mapping matrix is a interesting design, however, it would benefit from more analysis demonstrating that a simple, untrained matrix can perform the mapping effectively.

**Weakness**
- The method has limitation by design, the collbration across agents relies on transferring the whole KV cache across layers. this makes the colbrration very unlikely between models that are not exactly identical.
- The experiment result is not very persuasive, as the perfomrance gain are marginal aorss table 1,2,3,4. the standard deviations are needed for statistical signafacance.
- Missing a few strong test collbration baseline, e.g. the mixture of agent [1].

[1] https://arxiv.org/abs/2406.04692

---

> ### Author Rebuttal · Authors · 2026-03-30
>
> Dear Reviewer DCWk,
>
> Thanks for your insightful comments! We address your questions and concerns below.
>
> ---
> ## **[Q.1 & W.2] Standard Deviation in Tables 1,2**
> Thanks for this helpful question! We report standard deviations for all metrics in our experiments.
> |LatentMAS (Avg. Std.)|Acc|Token|Speed|
> |-|-|-|-|
> |Qwen3-4B|$\pm0.0026$|$\pm14.5$|$\pm8.2$|
> |Qwen3-8B|$\pm0.0043$|$\pm11.9$|$\pm13.8$|
> |Qwen3-14B|$\pm0.0019$|$\pm9.6$|$\pm21.2$|
>
> **Extended Reports:** We also provide more detailed std. analyses on LatentMAS and baselines. Please see tables in https://anonymous.4open.science/api/repo/LatentMAS_Submission-ED88/file/imgs/std.pdf?v=899338ba
>
> ---
> ## **[Q.2] Increased Repetitions on AIME 24/25**
> Thanks for this suggestion! We **increase the number of repetitions to 8 and 16**, and report the average accuracy below.
>
> - **8 Repetitions**
> |LatentMAS|Qwen3-8B|Qwen3-14B|Gain|
> |-|-|-|-|
> |AIME24|57.1|66.7|3.5|
> |AIME25|53.3|63.3|2.1|
> - **16 Repetitions**
> |LatentMAS|Qwen3-8B|Qwen3-14B|Gain|
> |-|-|-|-|
> |AIME24|57.3|66.5|3.4|
> |AIME25|53.5|63.3|2.2|
>
> **Key Findings:** The performance gain of LatentMAS remains consistent as we increase the repetitions.
>
> ---
> ## **[Q.3] Clarification on Qwen3 Reasoning Mode**
> Thanks for the question! In our implementation, we follow the default reasoning mode of Qwen3.
>
> For **intermediate agents**, we preserve the standard assistant prefix and the \<think\> tag, then perform latent thoughts generation.
>
> ```
> <|im_start|>assistant
> <think>
> Latent Thoughts Start...
> ```
> For **the final agent** outputing text answer, we follow the standard Qwen3 reasoning chat template.
>
> We have explicitly clarified this setup in our paper.
>
> ---
> ## **[Q.4] Effectiveness of the Alignment Matrix**
> Thanks for the suggestion! We provide a more intuitive explanation of the mapping matrix and conduct the comparison as requested.
>
> **(i) Intuition:** the alignment matrix projects last-layer embeddings back into the input space while preserving their semantic content.
>
> **(ii) New Comparison:** We further compare a single text model with a single latent model.
>
> **Setups:** For the single latent model, we first generate latent thoughts for fixed 40 latent steps, then switch back to standard LM-head decoding to produce final answers.
> |Acc. (on Qwen3-14B)|ARC-C|GSM8K|MedQA|MBPP+|
> |-|-|-|-|-|
> |Single Text Model|92.6|83.5|64.7|68.7|
> |**Single Latent Model**|**93.1**|**84.6**|**66.5**|**69.4**|
>
> **Key Findings:** The single latent model remains effective relative to the text baseline, verifying the efficacy of the alignment matrix.
>
> ---
> ## **[Q.5] "How do we draw Figure 5?"**
> Thanks for the question! We provide detailed descriptions below.
>
> In our implementation, we first collect 3 embedding vectors:
> - Original input embeddings $e_t$.
> - Last-layer embeddings $h_t$ before alignment.
> - Returned embeddings $e_{t+1}$ after alignment.
>
> Then we visualize how the unaligned $h_t$ drift away from the input embedding $e_t$, and how the aligned $e_{t+1}$ move back toward that space. Specifically:
> - **Density Plot (Figure 5 left):** We compute the PC1 value for each embedding vector and plot the distribution, with PC1 on the x-axis and probability density on the y-axis.
> - **Embedding Plot (Figure 5 Right):** We apply PCA to project each embedding vector into a two-dimensional space and directly plot the points for visualization.
>
> We have revised the description of Figure 5 in our paper for clarity.
>
> ---
> ## **[W.1] Collaboration on Non-identical Agents**
> Thanks for pointing this out! During method design, we primarily focus on the collaboration paradigm itself to enable training-free latent reasoning and communication.
>
> As discussed in `Appendix D`, extending the current paradigm to non-identical agents is an important direction to explore. **A natural and simple extension to support non-identical agents** is to train a lightweight adapter to align different hidden dimensions and map layers across models.
>
> We hope this helps the reviewer better understand the scope of our work.
>
> ---
> ## **[W.3] Comparison with MoA**
> Thanks for reminding this important work! We provide detailed justifications and new comparisons below.
>
> **(i) Rationale on Original Exps:** When comparing TextMAS, we control agent roles and collaboration style identical to LatentMAS. We didn't compare with Mixture-of-Agents initially, as its agent design (e.g., model types) is different from ours, making it hard to isolate the contribution of our collaboration design.
>
> **(ii) Adding New Baselines:** We further include two strong text-collaboration baselines, ReAct and MoA, using Qwen3-4B as the same backbone for all methods to ensure fair comparison.
>
> **Results:** Please see table in https://anonymous.4open.science/api/repo/LatentMAS_Submission-ED88/file/imgs/MoA.pdf?v=59476828
>
> LatentMAS performs favorably against new baselines and maintains efficiency gains.
>
> We have incorporated these comparisons into the paper and clearly cite and discuss these related methods.

---

> > ### Author Rebuttal · Reviewer_DCWk · 2026-04-02
> >
> > Thanks the author for the additional experiment and discussion. My concerns are all addressed

---

> > > ### Author Response · Authors · 2026-04-03
> > >
> > > Dear Reviewer DCWk,
> > >
> > > Thank you very much for your positive feedback! We are delighted that our rebuttal has addressed your concerns. We sincerely appreciate your insightful comments and have incorporated your suggestions to further strengthen the paper.
> > >
> > > Best regards,
> > >
> > > Authors of LatentMAS

---

### Decision · Program_Chairs · 2026-04-30

**Decision:**

Accept (spotlight)

**Comment:**

This paper introduces LatentMAS, a training-free framework that shifts multi-agent collaboration from text-based mediation to direct interaction within the continuous latent space. By generating auto-regressive latent thoughts from last-layer hidden states and exchanging them via a shared latent working memory (layer-wise KV caches), the framework enables lossless, efficient inter-agent communication without explicit decoding. Theoretical analysis demonstrates LatentMAS's superior expressiveness and lower complexity compared to standard text-based MAS, while extensive evaluations across nine benchmarks confirm substantial empirical gains: up to 14.6% higher accuracy, 70.8%–83.7% reduction in output tokens, and 4×–4.3× faster end-to-end inference. Reviewers unanimously recommend acceptance, praising the conceptual novelty, rigorous theoretical grounding, strong cross-domain empirical validation, and significant efficiency improvements. The work offers a principled, scalable paradigm for latent-space multi-agent reasoning that aligns well with ICML's emphasis on efficient and theoretically motivated machine learning systems.